

# Modelling landslide hazard under global change: the case of a Pyrenean valley

Séverine Bernardie[1], Rosalie Vandromme[1], Yannick Thiery[1], Thomas Houet[2], Marine Grémont, Florian Masson[1], Gilles Grandjean[1], Isabelle Bouroullec[3]

[1]BRGM, 3 avenue Claude Guillemin 45060 Orléans, France
[2]LETG-Rennes UMR 6554 CNRS, Place du Recteur Henri Le Moal, 35043 Rennes Cedex
[3]BRGM, 31000 Toulouse, France

*Correspondence to*: Séverine Bernardie (s.bernardie@brgm.fr)

**Abstract.**

Several studies have shown that global changes have important impacts in mountainous areas, since they affect natural hazards induced by hydro-meteorological events such as landslides.

To estimate the capacity of mountainous valleys to cope with landslide hazard under global change (climate change as well as climate- and human-induced land use change), it is necessary to evaluate the evolution of the different components that define this type of hazard: topography, geology and geotechnics, hydrogeology and land cover. The present study evaluates, through an innovative methodology, the influence of both vegetation cover and climate change on landslide hazard in a Pyrenean valley from the present to 2100.

Once the invariant features of the studied area, such as geology and topography, were set, we first focused on assessing future land use changes through the construction of four prospective socioeconomic scenarios and their projection to 2040 and 2100. These inputs were then used to spatially model land use and land cover (LUCC) information to produce multi-temporal LUCC maps. Then, climate change inputs were used to extract the water saturation of the uppermost layers, according to two greenhouse gas emissions scenarios. The impacts of land use and climate change based on these scenarios were then used to

modulate the hydro-mechanical model to compute the factor of safety (FoS) and the hazard levels over the considered area. The results demonstrate the influence of land use on slope stability through the presence and type of forest. The resulting changes are significant despite being small and dependent on future land use linked to the socioeconomic scenarios. In particular, a reduction in human activity results in an increase in slope stability; in contrast, an increase in anthropic activity leads to an opposite evolution in the region, with some reduction in slope stability.

Climate change may also have a significant impact in some areas because of the increase in the soil water content; the results indicate a reduction in the FoS in a large part of the study area, depending on the landslide typology considered. Therefore, even if future forest growth leads to slope stabilization, the evolution of the groundwater conditions will lead to destabilization.





These changes are not uniform over the area and are particularly significant under the most extreme climate scenario, RCP 8.5. Compared to the current period, the size of the area that is prone to deep landslides is higher in the future than the area prone to small landslides (both rotational and translational). On the other hand, the increase rate of areas prone to landslides is higher for the small landslide typology than for the deep landslide typology. Interestingly, the evolution of extreme events is related to the frequency of the highest water filling ratio. The results indicate that the occurrences of landslide hazards in the near future (2021 - 2050 period, scenario RCP 8.5) and far future (2071 - 2100 period, scenario RCP 8.5) are expected to increase by factors of 1.5 and 4, respectively.

## 1    Introduction

Global changes have impacts worldwide, but their effects are even more exacerbated in particularly vulnerable areas, such as mountainous regions. In these areas, a range of socioeconomic sectors (e.g., tourism, forest production, agro-pastoralism, natural resources) have experienced considerable change in the last two centuries, resulting in pressures on natural resources and traditions that are imposed by increasingly industrialized societies (Huber et al., 2005). Some mountainous regions have been extensively transformed, converting them from inaccessible and relatively poor areas into attractive destinations for the wealthy, and sometimes excluding long-time inhabitants from the economic benefits. In other cases, outmigration and an ageing population have led to economic declines in the agro-pastoral and forestry sectors.

Climate change will affect mountainous regions. An increase in the temperature in these areas has already been observed that is comparable to what has been observed in lowland regions (Kohler and Maselli, 2009). Some boundaries, such as the tree line, the limit of snow, and the limit indicating the presence of glaciers and permafrost, start to be modified. The hypothesis of global warming has now been validated by various studies (IPCC, 2007, 2014), even if the uncertainty of the climatic parameters is still up for debate. In this context, the snow and freezing lines will move higher in altitude. Precipitation may increase in the tropics as well as at to high latitudes, and may decrease in subtropical dry areas. Finally, the intensity of extreme meteorological events might also increase in the future. The climate evolution have significant impacts of natural hazards since most of them are induced by hydro-meteorological events, such as floods and different typologies of landslide (such as debris flows, permafrost rockfall instabilities, and erosion). For example, the IPCC notes that "There is high confidence that changes in heatwaves, glacial retreat, and/or permafrost degradation will affect slope instabilities in high mountains, and medium confidence that temperature-related changes will influence bedrock stability. There is also high confidence that changes in heavy precipitation will affect landslides in some regions" (IPCC, 2014). The vulnerability of mountainous areas to hydro-meteorologically induced hazards may increase in that way. In the Pyrenees, an OPCC report (2018) mentioned that "It is highly probable that the Pyrenees will see an increase in extreme weather phenomena". This may lead to more frequent floods, landslides, rockfalls and avalanches. Nevertheless, the quantification of the impacts of climate change on natural hazards related to hydro-geohazards remains a complex issue. Among hydro-geohazards, landslides are very sensitive to the hydro-meteorological conditions present in mountainous regions and are complex, which suggests that the analysis of the evolution





of the different mechanisms leading to landslides also remains complex. Gariano and Guzetti (2016) indicate that several landslide-triggering parameters may be affected by climate change. Indeed, climate change occurring in mountains may imply future modifications in temperature and precipitation patterns; this may lead to changes in the balance between snow, ice and rainfall, which ultimately will result in changes in precipitation quantity and seasonality. Landslides are also sensitive to the
presence of water within the layers that are susceptible to movement. As future trends in the climate may imply some modifications of meteorological parameters such as precipitation and temperature, the resulting underground water table level may evolve.

The sensitivity of landslides to climate change may depend on their typology, especially on the size and depth of the landslide (Crozier, 2010). As shallow landslides are generally governed by shorter-duration rainfalls, they may be more influenced by
the evolution of parameters in the short term parameters evolution, such as changes in the intensity of rainfall. In contrast, deep-seated landslides may be affected by long-term hydro-meteorological evolutions, such as changes in the monthly rainfall, seasonal snow, or groundwater. Several studies are related to the analysis of the impact of climate change on landslide occurrence. The review conducted on that topic, described in Gariano and Guzzetti (2016), indicates the increasing number of studies devoted to this issue. The future evolution of hydro-meteorological conditions implies that there will be some
modification in the frequency of landslide occurrence (Brunetti et al., 2012; Vennari et al., 2014; Gariano et al., 2015) and also in their seasonal temporality (Stoffel et al., 2014); these factors will evolve differently depending on the region of interest.

Landslide hazards may also be affected by global change with the evolution of the socioeconomic contexts, which may have some impacts on landslides through the evolution of vegetation cover. Indeed, vegetation influences deep-seated landslides very little, but this influence exists for shallow landslides and remains difficult to address. At the same time, the vegetation
cover increases the ground weight and thus tends to initiate ruptures, but it also increases the shear strength. The relative importance of these two contrary effects varies according to the localization of the vegetation cover on the slope. The stability is increased if the vegetation is present on the toe of the slope (Genet et al. 2010, Ji et al. 2012), but this stabilizing effect is reduced if the vegetation is located on the upper part of the slope (Norris 2008, Genet et al. 2010).

Many studies have been conducted based on historical observations; these analyses, which use empirical or numerical approaches, permit the analysis of the evolution of different features of landslides (frequency, typology, evolution of the mechanisms, etc.). The quantification of future scenarios is a key challenge in estimating the future trends in landslide activity with the future evolution of climate change, and estimating the uncertainties is necessary in these approaches. For this reason,
modelling constitutes an approach that is being increasingly developed in two different ways, depending on the size of the investigated area: a) physical models and b) statistical models. The majority of physical approaches are local and investigate a portion of a slope, a single slope, or a single landslide (Buma and Dehn, 1998, 2000; Dehn and Buma, 1999; Tacher and Bonnard, 2007; Bonnard et al., 2008; Comegna et al., 2013; Rianna et al., 2014; Villani et al. 2015, Collison et al. 2000, Chang and Chiang, 2011, Coe, 2012, Melchiorre and Frattini, 2012). In contrast, many studies based on statistical approaches analyse



future climate change impacts on landslides at regional scales (Jakob and lambert, 2009; Jomeli et al., 2009; Turkington et al., 2016; Ciabatta et al., 2016).

Recently, regional landslide susceptibility has been determined by considering climate change scenarios (Fan et al., 2013; Kim et al., 2014; Gassner et al., 2015; Shou and Yang, 2015; Wu et al., 2015), as well as regional landslide hazards (Baills et al.,

2013; Lee et al., 2014;Winter and Shearer, 2015). The advantages of such approaches are the quantification of the effect of future climate scenarios on landslides at a regional scale, permitting the consideration of the spatial variability of the environmental parameters. However, it appears that there are still few studies dedicated to the quantification of the impact of future climatic and land use scenarios on landslides.

In this study, the impacts of global change on landslide hazards is analysed according to two features: i) the evolution of land use, which can be analysed as the evolution of vegetation cover and its effects on landslides, and ii) the evolution of climatic conditions, which result in the evolution of the hydrogeological conditions. Different typologies of landslides are considered here, and the effects of global changes on each of them are analysed. This analysis is conducted within a Pyrenean valley in a high-elevation context. We first introduce an overview of the methodology for evaluating the impact of global change on

landslide hazards. Then, the mechanical and hydrogeological models are briefly presented, as well as the geological setting of the studied site. The socioeconomic and climatic scenarios and the way they are used in the mechanical and hydrogeological models are described. The last section presents the results of the modelled scenarios and a final discussion of the results.

## 2    Modelling methodology

The evaluation of the impacts of global change on landslide activities is realized through the analysis of the effects of both

vegetation cover and meteorological conditions on landslide activities; this analysis was conducted with the following methodology (Bernardie et al., 2017; Grandjean et al., 2018):

- The assessment of future land use is addressed through the construction of four prospective socioeconomic scenarios and their projection to 2040 and 2100, which are then spatially modelled with land use and cover change (LUCC) models;

- The climate change inputs for this study correspond to two greenhouse gas emissions scenarios. The simulations were

performed with the greenhouse gas (GHG) emissions scenarios (RCP: representative concentration pathways, according to the standards defined by the GIEC) RCP 4.5 and RCP 8.5.

The impacts of land use and climate change are then addressed through these different scenarios that provide the future evolution of hydro-mechanical parameter variations used to compute the hazard levels. The hydro-mechanical model is based on a large-scale slope stability assessment tool (ALICE) that combines a mechanical stability model, a vegetation module that

modulates the stability parameters to take into account the effects of vegetation on the mechanical soil conditions (cohesion and over-load), and a hydrogeological model (GARDENIA) that provides the water table level computed from meteorological temporal data. The main algorithm computes a spatially distributed FoS for the two selected future periods (2040 and 2100) and the different LUCC models.





## 2.1 ALICE slope stability model

ALICE® (Assessment of Landslide Induced by Climatic Events) was developed by the French Geological Survey (BRGM) to support landslide susceptibility assessment and mapping for areas ranging from individual slopes to regions (Sedan et al., 2013a; Vandromme et al., 2015; Thiery et al., 2017a, b; Vandromme et al., 2019, in review). Developed in a GIS environment,

it is a physically distributed model based on a limit equilibrium method that computes the factor of safety (FoS) along 2D profiles over the entire area. It can integrate different landslide geometries, the spatial and inherent heterogeneity of the surficial deposits and geology and their geotechnical parameters, triggering factors (i.e., water and seismicity) and land use change. The 3D geometry of the studied area is entered as a dataset in raster format: topography and layer interfaces are represented with different DEMs (digital elevation models). Geomechanical characteristics, namely, cohesion (c'), friction angle (φ') and

volumetric weight (γ'), are associated with each geologic and surficial deposit layer. These parameters can be implemented with a constant value or with probabilistic distributions to take into account the environmental variability of these parameters and the uncertainties associated with the values of these parameters (cf. Vandromme et al 2019, submitted). Like the other geometrical inputs of the model, the ground water table level (GWL) is implemented in raster format, as a piezometric map: a minimum level and a maximum level are required. Then, the GWL can be implemented empirically between a saturation level

from 0 (dry conditions, minimum level) to 1 (saturated conditions, maximum ground water level). The 2D profiles are automatically generated from a digital elevation model using the local drain direction (Vandromme et al 2019, submitted). A slope stability assessment is performed on each 2D profile, and each profile covers the whole studied area. The slope stability calculation is performed considering the forces applied on sliding bodies along a potential slip surface that can be circular or along an interface (cf. Vandromme et al 2019, in review). The rotational slip surfaces are characterized by length, a minimal

depth and a maximal depth. The shallow translational slip surfaces are defined by an interface, a length and a scarp angle. The shear strength of the soil along the potential failure plane is given by the Mohr-Coulomb failure criterion. The Morgenstern and Price (1965, 1967) method is used to calculate FoS (Vandromme et al 2019, in review). Once all parameters (geometrical, geotechnical, landslide type, triggering factor) are implemented in the model, several sliding surfaces are computed along each profile. The sliding surface with the lower FoS is kept, and this value is attributed to every cell included in the sliding surface.

## 2.2 GARDENIA hydrological model

The hydrogeological model GARDENIA (modèle Global À Réservoirs pour la simulation de DÉbits et de NIveaux Aquifères, Thiéry, 2003) is a lumped hydrological model. It simulates the main water cycle mechanisms in a catchment basin (rainfall, snowmelt, evapotranspiration, infiltration, runoff) by applying simplified laws of physics to flows through successive reservoirs. It considers a system of 3 tanks, reproducing the 3-layer characteristics of the hydrological behaviour of the soil: i)

the top zone, the first ten centimetres, where evapotranspiration occurs, ii) the unsaturated zone, where runoff occurs, and iii) the saturated zone. Once calibrated, the model permits us to i) calculate the balance of rainfall, snowmelt, evapotranspiration, run-off and infiltration into the underlying aquifer, ii) analyse the consistency between meteorological observations and





observations of flow rates or piezometric levels, iii) for a given catchment basin, recreate the flow rates of a river or spring and/or the piezometric levels at a given point in the aquifer during periods when measurements are not available, and iv) simulate flow rates resulting from dry periods or unusual precipitation sequences, groundwater piezometric levels, observed precipitation and precipitation resulting from scenarios (drought or high water periods resulting from climate change

scenarios). For our purpose, it allows the estimation of the daily local piezometric level evolution based on the meteorological parameters that might evolve due to climate change.

### 2.3 Linking slope stability and hydrological models

The spatialized piezometric level can then be integrated into the slope stability assessment equations developed in ALICE. In

this approach, a significant approximation is made by spatializing the piezometric level in the surficial formations; the ground reality is much more complex. This so-called piezometric level has to be considered an indicator of the water content of the soil, which will promote slope destabilization. This indicator is called the "water filling ratio": a value of one means that the water table level is at its maximum, and zero means that the level is at its minimum.

This method results in an analysis of the dynamic evolution of landslide susceptibility in the area of interest. Notably, run-out

is not accounted for in this study. Nevertheless, we still use the term "hazard" (Corominas et al. 2014) since this study analyses both the intensity and the occurrence of landslides.

### 3    The Cauterets basin, its geological setting and available data

The Cauterets municipality is representative of the climatic, lithological, geomorphological and land use conditions observed in the French Central Pyrenees (Viers, 1986). The municipality, an area of 157 km², is located in the intra-Pyrenean climatic

area (Barrère, 1952). It is highly affected by several natural hazards, such as landslides, rockfalls, debris flows and flash floods and was refurbished by RTM services at the end of the 19th century (De Crécy, 1988). This valley is one of the Pyrenean valleys with a north-south orientation. Dominated by ridgelines, with a maximum elevation of approximately 2500 m a.s.l., the municipality is subject to oceanic climatic influences from the west and to northwest flows. Thus, the average rainfall of 1157 mm.yr⁻¹ and the thermal amplitude of 13° are typical for a mountain area (Gruber, 1991). However, under the oceanic

influence, the valley is marked by a mountainous climate with intense storms during summer and autumn and large snowfalls during winter (it is often the snowiest resort area in France). This enclosed valley juxtaposes several types of fractured lithology (granite, limestone, schists, and sandstones) and reliefs with different exposures and strong elevation variations (Barrère et al., 1980). Consequently, the open slopes located above 1600 m a.s.l. with eastern exposure have more sunshine in summer and more snowfall in winter than the valley bottom or the slopes exposed to the west.

The test site is located in the middle part of the municipality of Cauterets in an area of approximately 70 km² and is characterized by a large variety of landslides. It can be subdivided into three geomorphological units. The southern unit is dominated by plutonic magmatic rocks (especially granites) overlapped by limestones, while the western and eastern units are





composed of metamorphic rocks (especially schists and sandstone) with intercalation of limestone. The lithology can be overlapped by morainic deposits on gentle slopes and morainic colluviums on the steepest slopes (Barrère et al., 1980; Viers, 1986). In more detail, each of these units can be described as follows:

1. The southern unit (ca. 18 km²) is drained by the Gave of Marcadau, the Gave of Gaube and the Gave of
Lutour. They all flow into the Gave of Cauterets. The lithology is fractured and shows its Hercynian heritage and Variscan and Alpine orogeneses (Majesté-Menjoulas et al., 1991; Majesté-Menjoulàs, 2018). In some locations, the valleys have kept the traces of the last glaciation and the last deglaciation with moraine deposits (sometimes very thick) plated on the slopes and fluvio-glacial deposits at the bottom of valleys (Barrère et al., 1980). The steep slopes (30-75°), which produce rockfalls, can be covered by screes. They were topographically modified and then covered with conifers at the end of the
19th century by RTM services (De Crécy, 1988).

2. The western unit (ca. 38 km²), principally drained by the Gave of Cambasque and the Catarrabes torrent, presents an irregular topography of alternating very steep scarps, steep convex slopes, planar slopes, and hummocky slopes. The southern part, dominated by the Soum de Grum and the Grand Barbat, is the domain of two old glaciers with very steep slopes that are carved in schists and partially covered by screes and lie below an irregular topography composed
of morainic deposits or weathered schists. Landslides in this area are located along streams and occur mainly at the point of contact with morainic deposits and weathered schists or are triggered in very weathered schists, as in the complex Lys landslide downstream of the ski resort located in the Cirque of Lys. To the north, the slopes are steeper and composed of schists and screes that are partially covered by open mixed forest and deciduous forest. This highly fractured area experiences rockfalls and large rock landslides, such as the large rockslide in the Arrouyes Valley, whose deposits are
located just upstream of the Limaçon glacial threshold (Barrère, 1952).

3. The eastern unit (ca. 14 km²), is characterized by very steep slopes (> 40°) carved in schists and sandstones, especially on the upper part of the slopes and in the northern basins. To the south, two ancient glacial cirques covered by grasslands constitute large, generally flat areas with irregular topographies between 1650 and 1750 m a.s.l. (the Lisey plateau and the Arriégeou). The former glacial cirque of Arriégeou is characterized by a large relict landslide probably
triggered during the last deglaciation, while the Lisey slope was one of the most dangerous streams and was subject to debris flows before work was done to improve its condition during the last century (Flurin, 2006). Several landslides, from shallow to deep complex landslides, have affected the unit. Complex deep landslides occur in highly weathered and fractured schist and sandstone along the steep banks of the torrents, while shallow landslides occur on the gentler slopes, where the contact between morainic colluvium and weathered lithology creates hydrological discontinuity that leads to
instability.

A landslide inventory was compiled at a 1:10,000 scale through a diachronic interpretation of several data sources (i.e. air-photo interpretation between 1950 and 2016, ortho-image analysis, landform analysis with an accurate DTM analysis and interpolation of triplets obtained by LIDAR in 2016, literature analysis) completed by field surveys. A total of 426 objects





were identified, including landslides with rotational shear surfaces, landslides with translational shear surfaces and deep-seated gravitational slope deformations. Each object was stored in a GIS database with its geometrical (perimeter, area, maximal length and width) and geomorphological characteristics (typology, magnitude and state of activity based on morphological features combined with the age of the event; Wieczoreck, 1984). A confidence mapping index (CMI) based on field

recognition, API and literature review (Thiery et al., 2007) is applied for each identified object. Of the objects identified, 83 % have a high confidence index score, 13 % have a moderate confidence index score and 4 % have a low confidence index score. This type of index is necessary for selecting recently or currently unstable objects and validating the failure simulations for recent and future periods. Because the goal of the study is to assess the probability of slope failure (i.e., initiation), the boundaries of active landslides are classified into two zones: (i) the landslide-triggering zone (LTZ) and (ii) the landslide

accumulation zone (LAZ).

Among the different inventoried objects, only landslides with translational and rotational shear surfaces were selected to be modelled with ALICE. Deep-seated gravitational slope deformations are slow processes with a particular morphostructure, and occur in different conditions from those of other landslides (Crosta et al., 2013). Fig 1 and Table 1 depict the morphology

and morphometric/environmental characteristics of the different landslide types that have high confidence index values. Translational landslides have a nearly planar shear surface primarily located in the shallow part of the slope, whereas rotational landslides have a circular shear surface that occurs at different depths (Dikau et al., 1996).

**Table 1 – Characteristics of the different landslides**

| Landslide type | Number (n) | Depth (m) | Mean slope (°) | Mean size (m²) | Materials involved |
|---|---|---|---|---|---|
| Shallow translational | 225 | ≤ 2 | 35 | 2,512 | Colluviums and weathered materials |
| Shallow rotational | 57 | ≤ 2 | 32 | 4,200 | Moraine deposits (colluvium, removed moraines, moraines in place) |
| Moderately deep rotational | 8 | ≤ 6 | 24 | 16,695 | Weathered schist and sandstone and moraine deposits |
| Deep rotational | 56 | > 6 | 22 | 14,062 | Weathered schist and sandstone and moraine deposits |

We will consider the following four typologies: i) shallow translational landslides, which occur mainly in various colluviums and weathered materials (especially in weathered schist) from gentle to very steep slopes (from 12° to 47°); ii) shallow rotational landslides, which occur in thin moraine deposits (moraine colluviums, removed moraine deposits or moraines in place) from moderately to very steep slopes (from 17° to 50°). The majority of these are located near streams and are triggered




by a basal incision caused by torrents; iii) moderately deep and iv) deep landslides, which have a common rotational shear surface in the initiation zone. These occur in materials such as moraine deposits and/or weathered materials at deeper depths. They are located on gentler slopes than shallow landslides (from 11° to 35°).

Table 2 outlines the main predisposing factors affecting each landslide type. The different thematic data are derived from (i) air-photo interpretation analysis, (ii) airborne LIDAR surveys, (iii) satellite imagery analyses, and (iv) field surveys. The DTM (10 m resolution) is derived from the airborne LIDAR data. The slope gradient map and the flow accumulation map are derived from the DTM. The lithological map is based on the main lithological units described in a geological map produced by the French Geological Survey (Barrère et al., 1980) at a 1:50,000 scale and is completed by fieldwork. The surficial formation

map is obtained by a geomorphological approach with the segmentation of homogeneous areas of the landscape associating surficial formation type, their facies, and the processes following the approach developed by Thiery (2007). The surficial formation thickness map is obtained from field observations of surficial formation outcrops along the stream and the slopes. The thickness of the formations is closely associated with the slope degree. For instance, screes are located on steep slopes, whereas moraine colluviums are located on gentler slopes. Therefore, an exponential regression function obtained by plotting

the thicknesses and the slope degrees derived from the DTM is computed to obtain a spatial prediction and continuous values for each type of surficial formation (Thiery et al., 2017b).

**Table 2 Main predisposing factors for each landslide**

| Factor | Subfactor | Source of information and approach used to obtain data for this study |
|---|---|---|
| Landslides | Shallow rotational Shallow translational Moderately rotational Deep complex with rotational upper part | Field survey and geomorphological approach |
| Topography & derivatives | Topography | LIDAR data |
| | Slope | Derived from the topography |
| | Flow direction | Derived from the topography and slope (with an 8-direction flow model) |
| | Flow accumulation | Derived from the Flow direction |
| Geology | Lithology | Defined geological map (1: 50 000, Barrère et al., 1980) completed by field observations |
| | Surficial formations | Defined by field observations |
| | Thickness of different surficial formations | Defined by field observation and the exponential relationship between thickness and slope |





| Geotechnical parameters | Cohesion | Defined using related literature based on field investigations |
|---|---|---|
| | Angle of friction | |
| | Bulk unit weight | |
| Triggering factors | GWL | Defined by hydrogeological modelling (GARDENIA®) computed with rainfall data and/or climate change scenarios |
| Land use | 4 land use scenarios | Defined by remote sensing analysis and prospective scenarios (LLUCC) |

**Fig 1 - Landslide location and overview of the main variables introduced in the ALICE model : a) geographical location; b) landslide inventory; c) Degree slope; d) surficial layers; e) Thickness of surficial layer; f) land use**





## 4    Scenarios

### 4.1 Socioeconomic scenarios and LUCC mechanical effects

The overall methodological approach for constructing socioeconomic scenarios is fully described in Houet et al. (2017). It consists of working with stakeholders to co-construct fine-scale socioeconomic scenarios based on existing national or regional

sector pathways to produce spatially explicit local LUCC maps. The method relies on two participatory workshops aiming first, to define the narrative and second, to validate the narrative and pre-identify the areas of future land use change. Meanwhile, the use of the LUCC model allows the simulation of the land cover changes induced by the future land use changes which, in turn, can have feedback effects on land cover. Thus, the narrative scenarios are defined to produce relevant inputs for the LUCC model, while the model itself is developed to be able to represent the likely land cover changes identified in the

narrative scenarios and to provide quantitative outcomes to illustrate the narratives. Four scenarios have been defined according to this methodology, resulting in future land cover maps for the four scenarios in 2040 and 2100 (Fig 2):

-    Abandonment of the territory ;

-    Sheeps and woods ;

-    A renowned tourism resort ;

-    The green town.

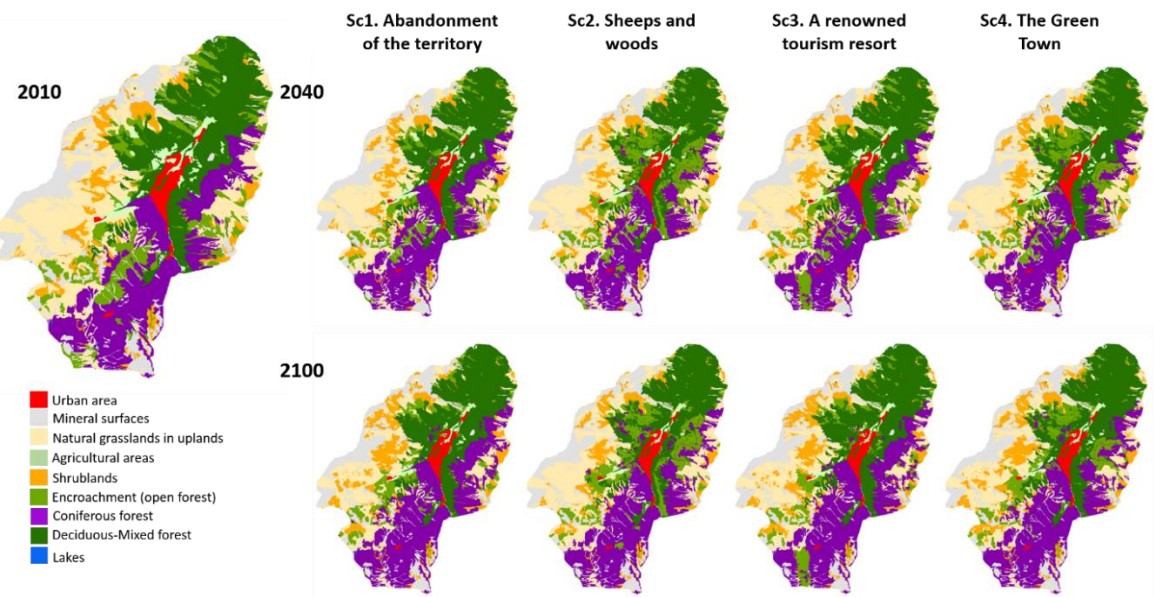

**Fig 2 - Maps of existing and future land cover for the four scenarios in 2040 and 2100 (Houet et al., 2017)**

Land use can have a significant influence on slope stability through different processes: i) the vegetation can reinforce the

surficial stability with reinforcement from roots; ii) the vegetation adds weight to the slope; and iii) the vegetation modifies





the hydraulic system. We state here that a single mechanical action can encompass these processes by considering the contribution of the cohesion offered by the roots, which is the predominant effect of vegetation on slope stability (Kokutse and al., 2016, Norris et al., 2008). The influence of the roots is therefore taken into account as an additional cohesion value. As roots can vary according to their orientation, most studies consider their influence on the friction angle of the material to be

negligible (Norris et al., 2008). Different models can be found for estimating cohesion; their quality varies and requires several types of biological input data (root resistance, root density and root morphology). Site effects are also significant, making it impossible to find a single value for cohesion for each species in the literature. When the site conditions are not limiting, tree root systems are classically categorized into three types: taproots, heart roots and flat roots. Wu et al. (2004) investigated how the three root morphologies lead to breaks during a slide and found that pivoting systems are likely to mobilize the full potential

of root resistance because the stresses are concentrated on the main root. On the other hand, for fasciculate and plate systems, the stresses are distributed over the set of roots that are thinner and yield one after the other, which does not mobilize the full resistance potential of resistance of the system.

In the context of Cauterets, the presence or absence of forest constitutes the main parameter that may modify the slope stability. This study considers 4 land cover types that may have significant influence on slope stability: a) no trees (urban areas,

scrublands, natural grasslands, agricultural areas, mineral surfaces); b) coniferous forest; c) deciduous forest; and d) encroachment (open forest).

Table 3 summarizes the additional cohesion provided by each type of forest considered in the study area.

**Table 3 - Additional cohesion from each type of forest**

| Classification according to Houet et al., 2017 | Principal type of forest in the area | Additional cohesion (kPa) |
|---|---|---|
| No forest | No forest | 0 |
| Coniferous forest | Coniferous | 8 |
| Deciduous forest | Deciduous | 12 |
| Recolonization of the forest (open and sparse) | Open | 5 |

The land uses in the current and future land use maps were categorized according to these classifications.

The depth of the influence of vegetation is limited by the maximum extension of the roots; Bischetti et al. (2009) quote Schichtl (1980), who estimates that in mountainous terrain, the roots do not exceed 1 m in depth. This value is comparable with those

indicated by Crow (2005), who found that 90 to 99 % of the root mass is less than 1 m deep. In this study, we considered the influence of vegetation to be limited to 1 m below the DTM.

Fig 3 shows the changes in the area of the four categories of land use (coniferous, deciduous, open forest, and no forest) for 2040 and 2100 in the 4th scenario; in 2010, a majority of the area was composed of forest (52 %), with a balance between coniferous (19.3 %) and deciduous forests (21.7 %) and a smaller open forest area (11 %). The future changes show a slight





increase in the forested area in 2040 and a larger increase in 2100, regardless of the scenario. Considering the proportions of the different types of forest, the most important changes concern the "abandonment of the territory" and "sheeps & woods" scenarios. There is a decrease in open forest in the "abandonment of the territory" scenario, whereas open forest increases for the "sheeps & woods" scenario; in the scenario "green town", the proportions of the different land uses in 2040 remain similar

to those in 2010, and the coniferous forest surface increases in 2100. Finally, the "renowned tourism resort" scenario shows an increase in the area of coniferous and deciduous forests.

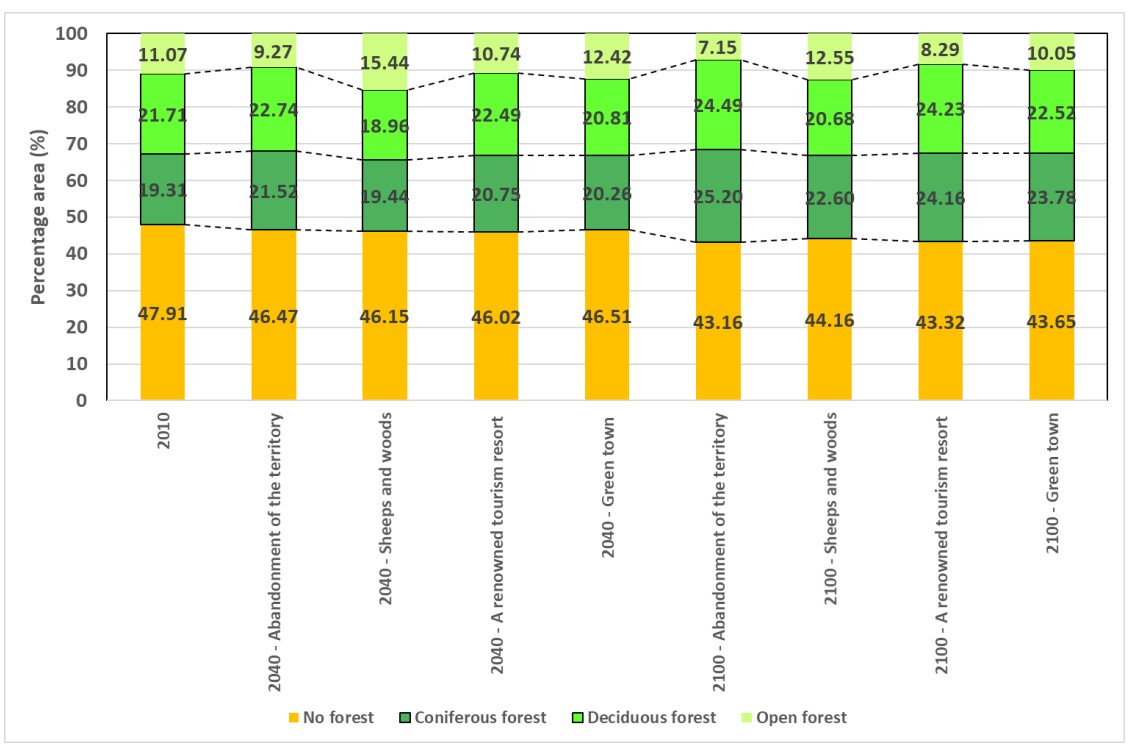

**Fig 3 - Evolution of land use in 2010, 2040 and 2100 for the four scenarios**

### 4.2  Climate change scenarios and hydrogeological inferences

To take climate change into account, we selected two contrasting greenhouse gas emissions scenarios, labelled on DRIAS

(2014) as RCP 4.5 and RCP 8.5 and computed using the ALADIN-Climate model of Météo-France.

Three periods were defined that are considered to be representative of the current year, 2050 and 2100: i) Reference period, 1981 – 2010; ii) Short-term period, 2021 – 2050; and iii) Long-term period, 2071 – 2100. A 30-year duration is commonly used for meteorological parameters (e.g., precipitation) and defined according to WMO criteria (World Meteorological Organization, 2019).





From a general point of view, the climate models show a tendency toward an increase in extreme precipitation events and annual cumulative precipitation in the short and long term. The model results depend on the elevation of the area of interest. In the highest areas, the models show an increase in cumulative precipitation. At the lowest points, they indicate a slight increase in the short term and a small decrease in the long term. Concerning the temperature, the models clearly indicate a

significant increase in the temperatures in the short (+1.5°C) and long term (+ 4°C), resulting in large changes in the precipitation pattern (the balance between snow and rainfall). Fig 4 shows that both models indicate the same tendency in the short term, but they show some significant differences in the long term. Moreover, there is a strong increase in liquid precipitation in the long-term RCP 8.5 scenario, coupled with a decrease in solid precipitation over the same period. For the long-term RCP 4.5scenario, liquid precipitation increases slightly compared to that in the short term. The solid precipitation

also decreases for the same period.

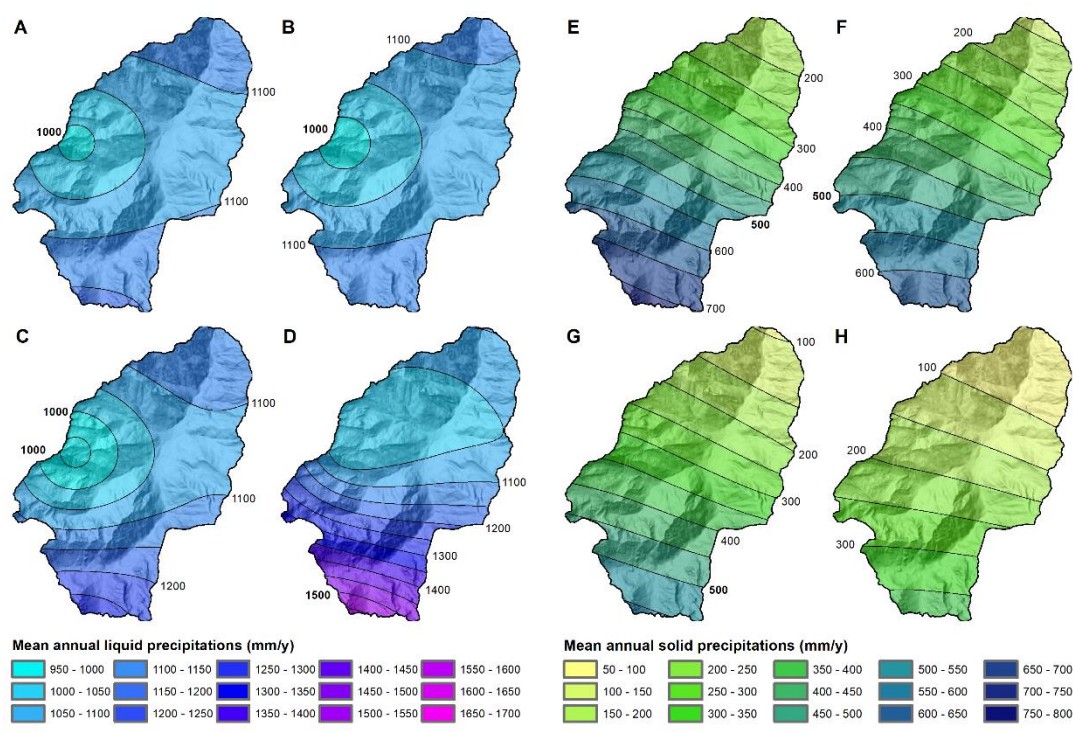

**Fig 4 - Mean annual liquid and solid precipitation in the short and long term for RCP4.5 and RCP8.5 scenarios (from DRIAS, 2014). For liquid precipitation : a) 2021-2050 period, RCP4.5 scenario ; b) 2021-2050 period, RCP8.5 scenario ; c) 2071-2100 period, RCP4.5 scenario; d) 2071- 2100 period, RCP8.5 scenario. For solid precipitation : e) 2021-2050 period, RCP4.5 scenario ; f) 2021-2050 period, RCP8.5 scenario ; g) 2071- 2100 period, RCP4.5 scenario; h) 2071- 2100 period, RCP8.5 scenario.**

With the GARDENIA model, which transforms precipitation rates into water table variations, the daily water table level is computed from the daily meteorological parameters. In this way, it is possible to obtain the distribution of the water table level





between low and high piezometric levels that reflect the minimum and maximum indicators of the soil water content (i.e., filling ratio). The piezometric level series, when converted into the water filling ratio, allows calculations to determine water level classes (WLC) and the associated frequency of occurrence, $f(WLC)$ for each WLC. Fig 5 shows the frequency distribution of the WLC of water level filling ratios for the current period (1981-2010), two future periods (2021-2050 and

5   2071-2100), and 2 climatic scenarios (RCP4.5 and RCP 8.5). This figure demonstrates the significant increase in the mean water table level in future periods, especially between 2071 and 2100 under the most extreme scenario (RCP8.5).

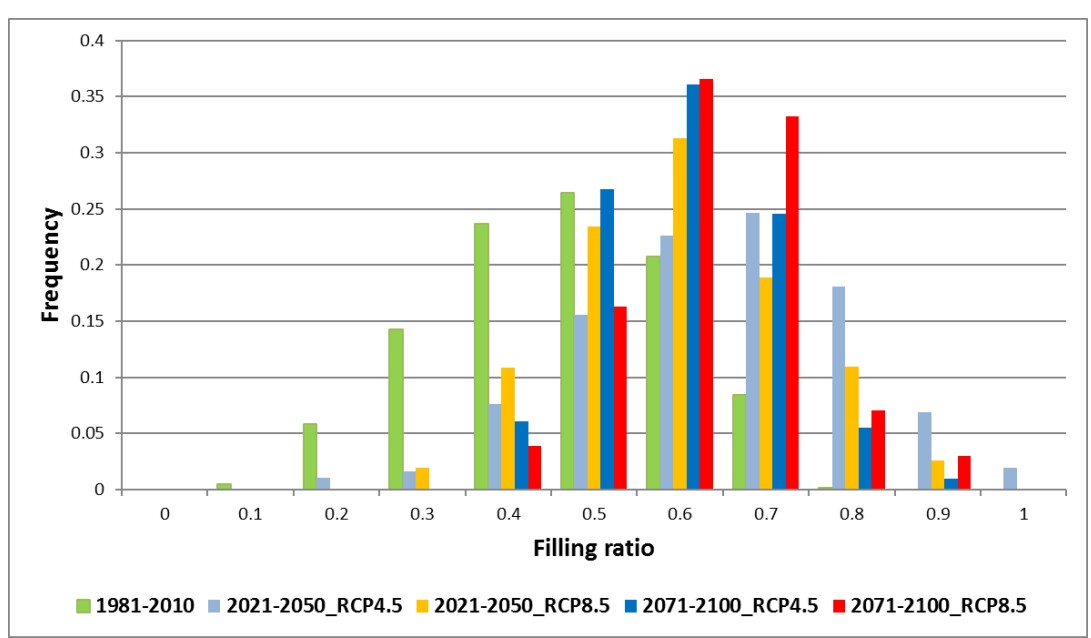

**Fig 5 - Frequency of the water table filling ratio**

## 5   Results

### 5.1   Landslide hazard in 2010

For each landslide type, 10 simulations were performed according to the 10 ground water filling ratios (from 0 to 1). Fig 6

15   depicts the surface occupied by the different classes of hazard (based on FoS values according to Vandromme et al., submitted). As seen in Fig 8, this analysis shows that the highly susceptible surface areas are the largest for large landslides and the smallest for small landslides.

The relationship of the evolution of the hazard to the water table level depends on the type of landslide.





For the translational typology, the graph indicates that the high hazard class area is insignificant until a 0.6 water filling ratio is reached; the hazard is then highly amplified with a strong increase in the area of very high and high landslide hazard areas. For small rotational landslides, the area susceptible to landslides is very low at lower water filling ratios until the 0.5 value, when the area prone to landslides increases slightly. The area of the high and very high hazard classes remains low, even at

the highest water filling ratio. At a water filling ratio of 0.8, there is a change in the evolution of the very high and high hazard classes. In contrast, the behaviour of medium and large landslides indicates a larger high hazard area : indeed, for the medium rotational landslides, the area that is susceptible to landslide remains higher than that for the small rotational landslides. The area prone to landslides increases linearly with the lower water filling ratio values (from 0 to 0.7 for the very high and high hazard classes; from 0 to 0.3 if the medium class is included). The area then increases greatly at higher water filling ratios.

Finally, when considering large landslides, a similar pattern is observed as for medium landslides, with a linear increase with lower water filling ratios (from 0 to 0.5 for the very high and high hazard classes; from 0 to 0.2 if the medium class is included ); the surface prone to landslides then rises substantially at higher water filling ratios.

In summary, a threshold of behaviour appears in the results, changing from low hazard levels and linear evolution to higher hazard levels and exponential evolution with the water filling ratio; this threshold depends on the typology of the landslide.

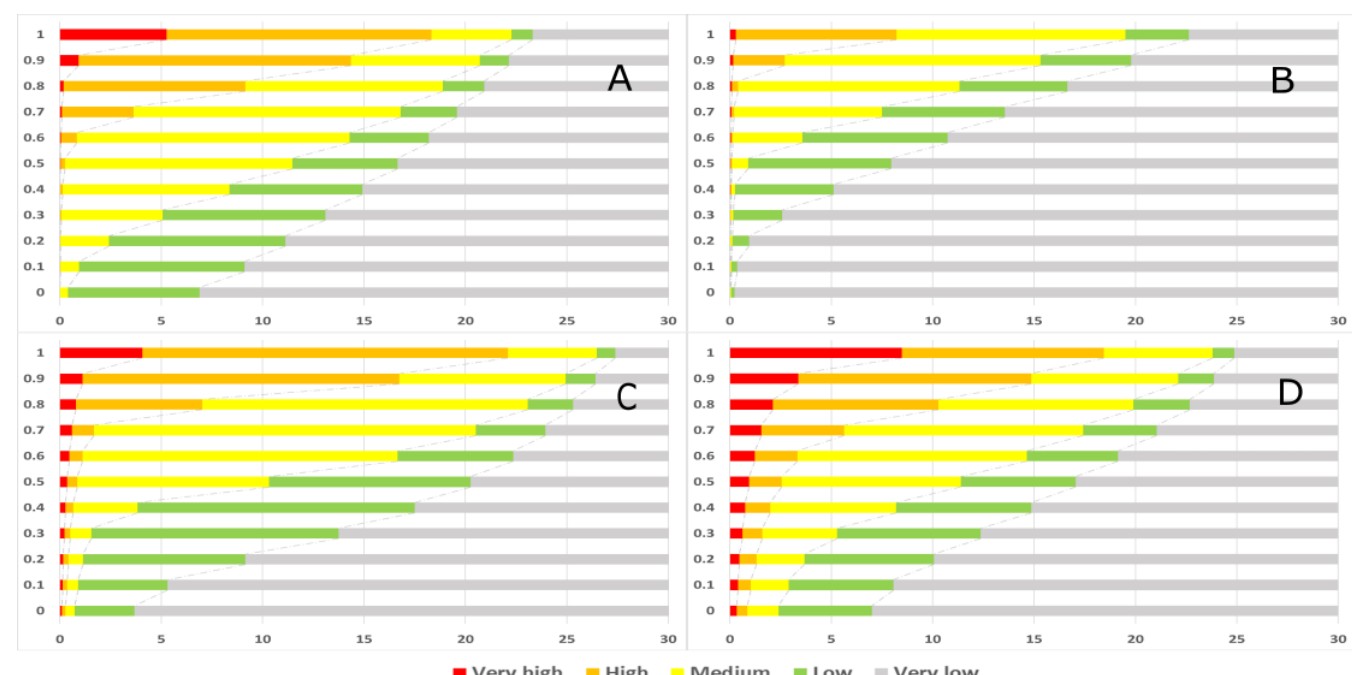

**Fig 6 – Evolution of the hazard level according to the water table level for the 4 types of landslide (2010) A) translational B) small rotational C) medium rotational, D) large rotational**

Figs 7 and 8 illustrate landslide hazard maps for a conservative scenario based on a GWL of 0.9 to explain the landslide events observed in 1992 over the ski resort (Fig 1B2) and over the Grange de Pan (Fig 1B1). This type of situation, which corresponds





to slope destabilization, is currently observed in moraine deposits overlapping weathering materials either in the French Pyrenees (Lebourg et al., 2003a) or in the French Southern Alps with similar geomorphological conditions (Thiery et al., 2017, Vandromme et al., submitted). Each map shows the initiation areas of the different inventoried active landslides. From the LTZ and the very high and high hazard classes defined from the FoS values, several statistical tests were computed (i.e.,

relative error = ξ; receiving operating characteristic curve and associated area under curve = ROC (AUC); prediction rate = PR (AUC); true positive rate = TPr; false positive rate = FPr; Fig 8). These classical evaluation and validation tests allow the assessment of the predictive power of the simulated maps (Brenning, 2005). The relative error (ξ) measures the model performance; it is defined as the total proportion of observations (i.e., the LTZ) correctly classified in the very high and high hazard classes (Thiery et al., 2007). The ROC (AUC) indicates the degree to which the LTZ is explained by the computed FoS

(hazard value). The ROC (AUC) is a tool for validating the different models by a threshold-independent measure of discrimination between the proportion of correctly predicted positive cases and the proportion of correctly predicted negative cases (Brenning, 2005). The values fluctuate between 0.5 (no discrimination) and 1 (perfect discrimination). The PR (AUC) is computed on the basis of the different FoS values and the LTZ. It is obtained by varying the decision threshold and plotting the respective sensitivities against the total proportions of the LTZ. It provides a validation index of the simulated maps (Chung

and Fabbri, 2003). Finally, the true positive rate and the false positive rate are the proportion of correctly classified observations and the proportion of incorrectly classified observations, respectively (Thiery et al., 2007; Vandromme et al., in review).

For shallow translational landslides, the relative error is low (i.e., ξ = 0.21), and the true predictive rate is high (i.e., TPr = 0.80), indicating that LTZs are well recognized by the model when the water table is high. This situation corresponds to the different field observations. For shallow rotational landslides, the simulated map does not recognize current unstable surfaces very well; the relative error is high; ξ = 0.79. However, the ROC (AUC) and PR (AUC) indices show high values. This is

partly due to the low surface area of this type of landslide. For moderately deep rotational landslides, the LTZs are well simulated by the model (i.e., ξ = 0.37 and TPr = 66 %). Finally, for deep rotational landslides, the relative error is relatively low (i.e., ξ = 0.35) and the ROC (AUC) and the prediction rate are high (i.e., ROC (AUC) = 0.82; PR (AUC) = 0.75). This means that the models with such a water table level are able to simulate this particular type of landslide, which has a deep

shear surface, well and that the models are discriminating.

**Fig 7 - Landslide hazard map for 2010 a) shallow translational; b) shallow rotational; c) moderately deep rotational; d) deep rotational**




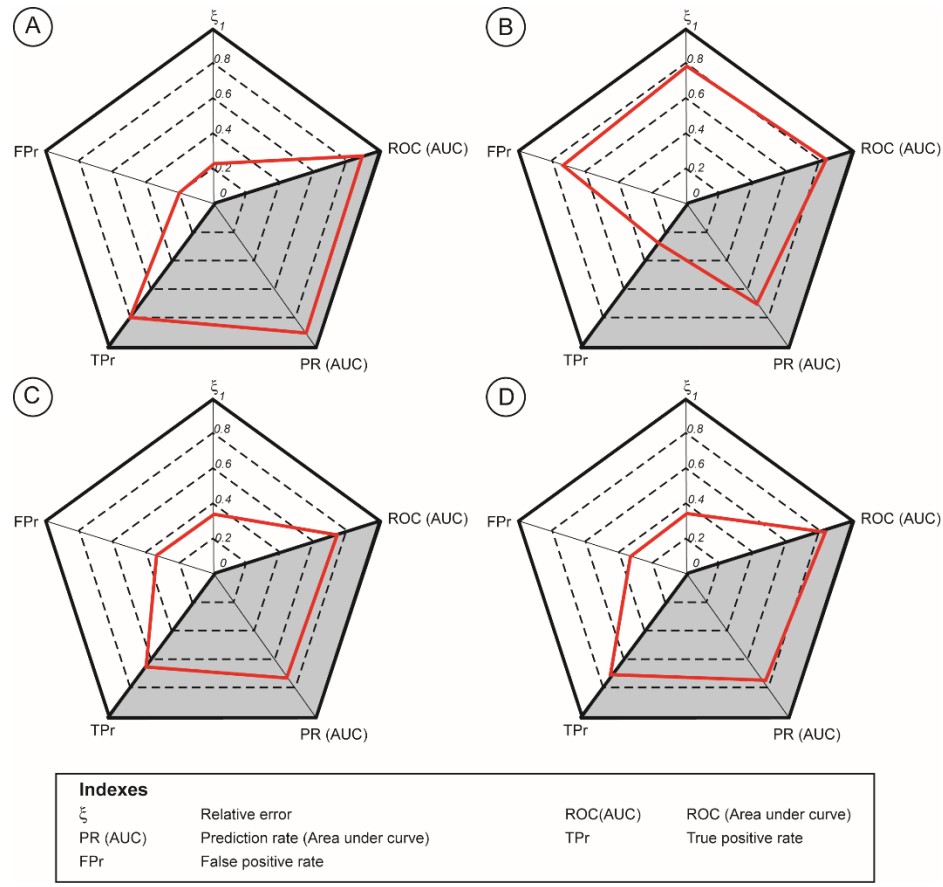

**Fig 8 - Statistic indicators between initiation areas and the very high and high hazard classes for A) shallow translational; B) shallow rotational; C) moderately deep rotational; D) deep rotational**

## 5.2 Future evolution of landslide hazard

### 5.2.1 Impacts of future land use on landslide hazard

Fig 9 shows maps of the differences in FoS values between the current and future periods for the four socioeconomic scenarios and the 2 future periods, with the objective of specifically analysing the influence of future land use on landslide hazards. Climate conditions are constant and set to the actual conditions. This analysis was performed on small rotational landslides; this landslide typology might be sensitive to the evolution of land use. In contrast, the stability of deeper landslides is influenced by the effects of vegetation.

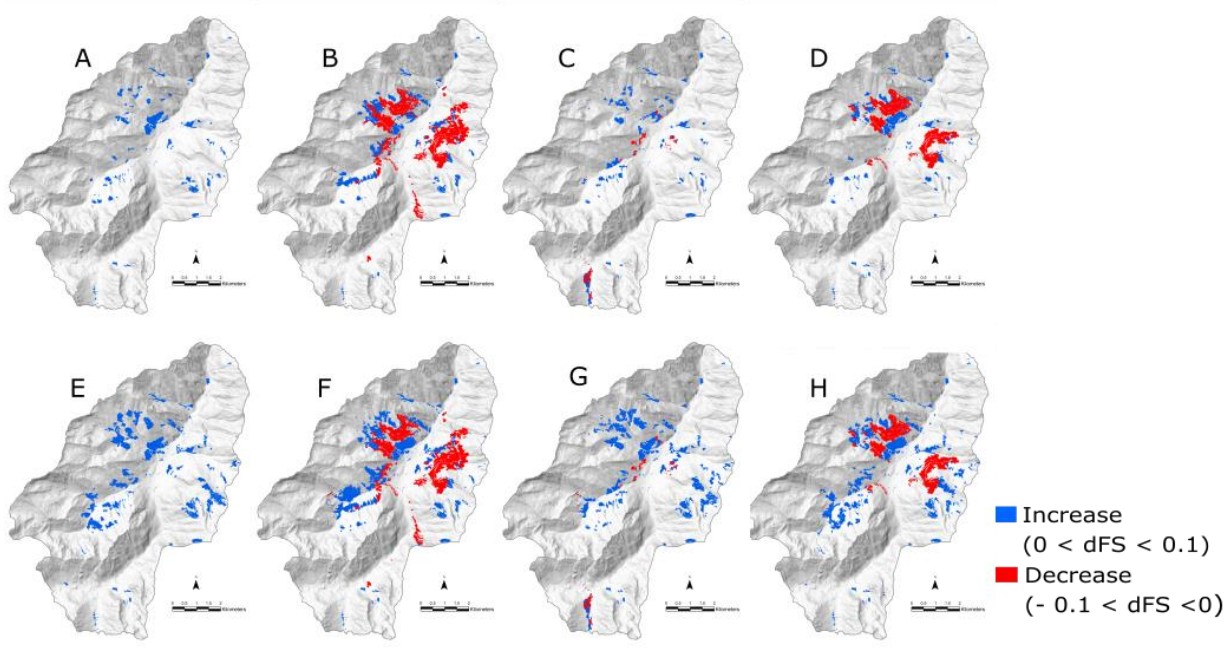

**Fig 9 - Map of differences in FoS between current and future periods for the 4 scenarios and the 2 future periods: A) abandonment of the area in 2040; B) sheep and woods in 2040; C) a renowned tourism resort in 2040; D) a green town in 2040; E) abandonment of the area in 2100; F) sheep and woods in 2100; G) a renowned tourism resort in 2100; H) a green town in 2100. The water table is constant.**

These maps indicate some differences between the current and future landslide hazards. These differences are characterized by local increases and decreases in slope stability. These confined contrasts are significant, although they remain limited, as the change in the FoS ranges from -0.1 to +0.1, which is quite small.

When considering the impacts of future land use on hazard level changes in detail, we observe the following:

In the "abandonment of the territory" scenario, we observe a general increase in slope stability, which is in good accordance with the reduction in open forest highlighted in Fig 4. In the same way, the "a renowned tourism resort" scenario is characterized by an increase in stability, linked with the increase in deciduous and coniferous forests and very limited areas of decreasing slope stability. In contrast, the "Sheeps and woods" scenario results in the destabilization of some areas because of the development of open forest. This scenario shows contrasting spatial evolution, as other areas become more stable. Finally, the "Green town" scenario results in more localized destabilization areas than the "sheep and woods" scenario, with patches of open forests and some improvement in stability in areas where coniferous forests are predicted to increase.

5.2.2    Impacts of future climate on landslide hazard





We integrated climate change into the analysis of future landslide hazards. Various analyses can be performed on the results. In this paper, one of the indicators that we intend to quantify is the tendency of future hazard evolution. To this end, we computed a "mean" of future landslide hazards. We computed the $FoS_{MEAN}$ indicator, which is the mean FoS value for each cell in the studied area:

$$FoS_{MEAN} = \sum_{WLC=1}^{n} [f(WLC) \cdot FoS(WLC)]$$

where WLC is the water level class and $f(WLC)$ is the frequency of having the filling ratio equal to the WLC, as determined in Fig 5. "n" is the number of filling ratio level classes; in this case, it is equal to 11 (see Fig 5). In other words, we summed the hazard maps corresponding to the water table ratio, weighted according to the distribution of the water level filling ratio

10 classes for the analysed period shown in Fig 5. The final result of the process is a single map that represents the mean landslide hazard over a given period.

Figs 10 and 11 show maps of the difference in the FoS between the future (long term) and current periods for the "small rotational" and "deep rotational" landscape types under the two climate scenarios, RCP4.5 and RCP8.5. In these figures, only

15 the "abandonment of the area" socioeconomic scenario is only considered.
Regardless of the scenario considered, there is a decrease in slope stability. The FoS change ranges between -0.1 and -0.2. Therefore, even in a case where the future evolution of the forest stabilizes the slopes, the evolution of the groundwater table destabilizes the area more.

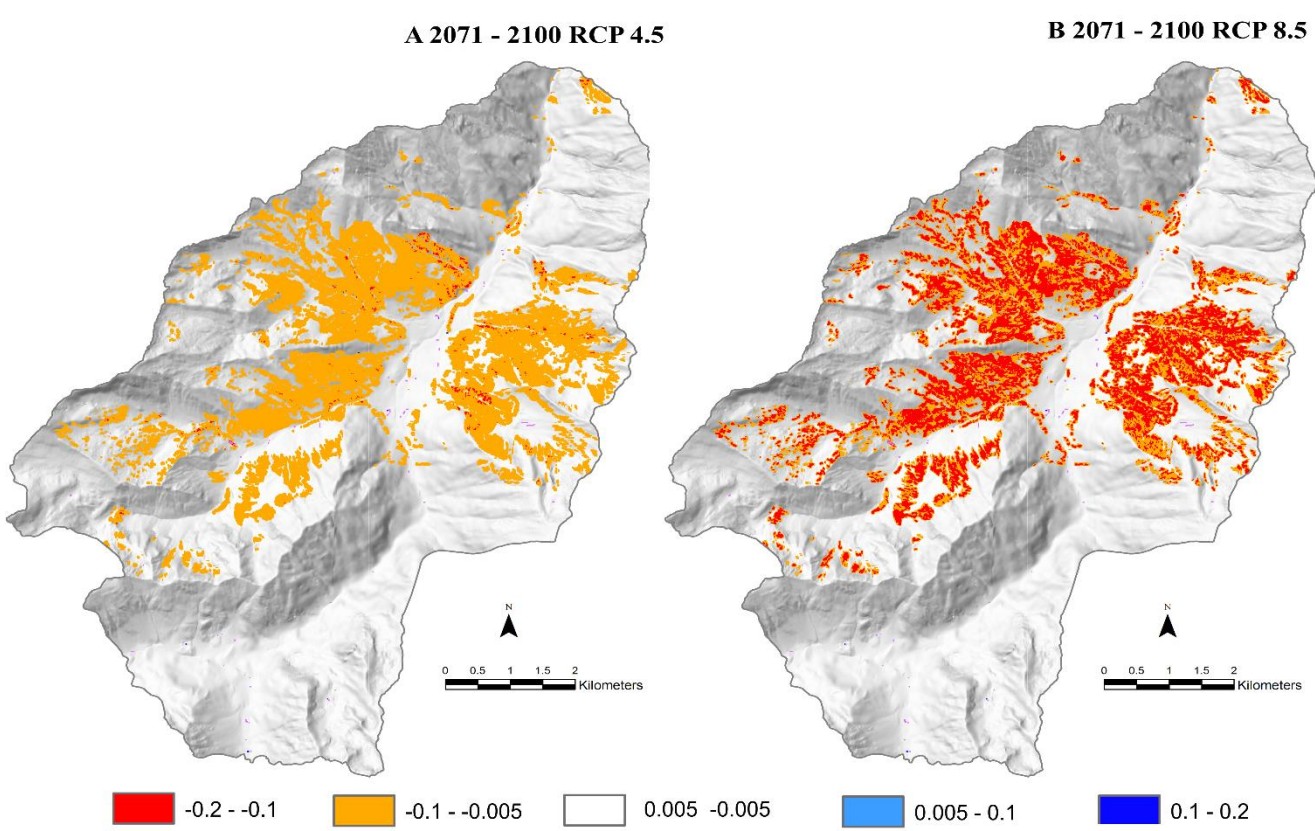

**Fig 10 – Maps of the difference in FoS between the future and current periods – Shallow rotational landslide**

**A 2071 - 2100 RCP 4.5**                     **B 2071 - 2100 RCP 8.5**

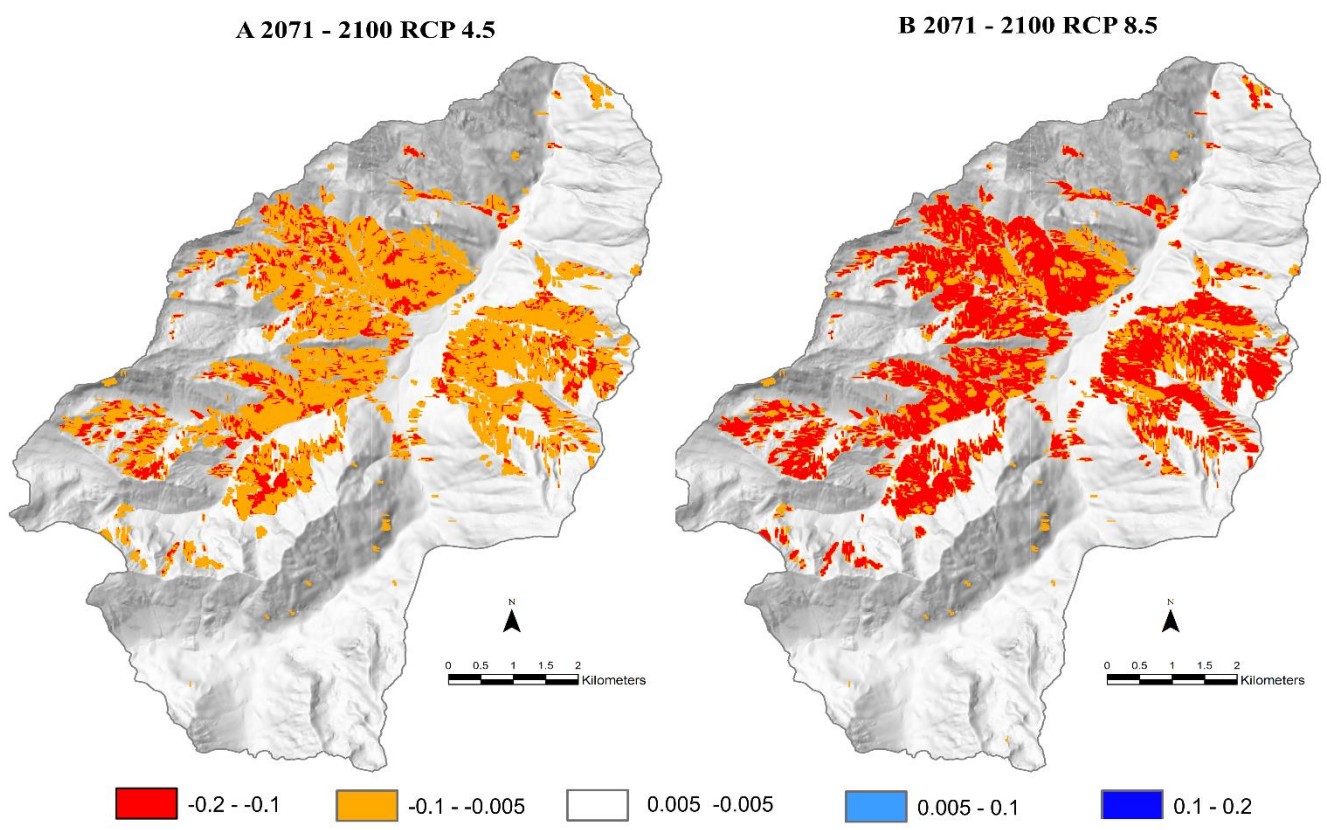

| | | | | |
|---|---|---|---|---|
| 🟥 -0.2 - -0.1 | 🟧 -0.1 - -0.005 | ⬜ 0.005  -0.005 | 🟦 0.005 - 0.1 | 🟦 0.1 - 0.2 |

**Fig 11 – Maps of the difference in FoS between the future and current periods – Deep rotational landslide**

These figures can be more precisely analysed, as seen in Fig 12, which provides the surface percentage for each hazard class
in the whole study area for each typology of landslide, each climatic scenario, and the two future periods. For clarity, the area
corresponding to the very low hazard class has been removed from the figures.

For the small rotational landslides, there is a strong increase of the medium hazard class, from 0.29 % to 3.23 % in 2040 for
the RCP 4.5 scenario; the high and very high classes remain at the same very low level. Thus, the sum of the surface classified
in the three highest levels increases from 0.34 % in 2010 to 2.12 % in 2040 (RCP8.5) and 2.78 % in 2100 (RCP8.5).

For the translational landslides, the most important increase occurs in the high hazard class, whereas there is a decrease in the
low hazard class, indicating that there is a transfer from low to medium and high classes. The total area classified in the three
highest levels is increases from 5.95 % in 2010 to 8.04 % in 2040 (RCP8.5) and 8.49 % in 2100 (RCP8.5). The very high class
remains at a very low level.

For the moderately deep landslides, some significant changes are evident, with a large area of the low hazard class changing
to the medium hazard class. The total area classified in the three highest levels increases from 4.83 % in 2010 to 9.48 % in
2040 (RCP8.5) and 10.27 % in 2100 (RCP8.5).



The deep landslides evolve differently, as the medium, high and very high hazard levels increase significantly, from 5.94 % in 2010 to 8.81 in 2040 (RCP8.5) and 8.84 % in 2100 (RCP8.5).

**Fig 12 - Evolution of the areas of the four highest hazard classes according to future CC scenarios – Abandonment of the territory scenario – A - small translational landslide; B - small rotational landslide ; C - moderately deep rotational landslide ; D - large rotational landslide**

## 6    Discussion

10      This study analyses the evolution of future landslide hazard under the effects of land use and climate changes. An accurate map provides a reference for establishing a comparison with the current state of landslide hazards. The criteria computed to quantify the accuracy of the model permit the validation of the model. For the four landslide types, the different models computed for 2010 are considered representative of this Pyrenean mountainous context. The triggering conditions used in the





model are similar to those observed in the field and proposed by Fabre et al. (2002) and Lebourg et al. (2003b). The statistical results are satisfactory for each landslide typology (e.g., the relative error $\xi = 0.21$ for the shallow translational, $\xi = 0.79$ for the shallow rotational, $\xi = 0.37$ for the moderately deep rotational, and $\xi = 0.35$ for the deep rotational). Even if the $\xi$ for shallow rotational landslides is slightly lower, we consider the results valid, as the majority of these landslides occur near

torrents due to basal incision from the torrents. Hence, ALICE® is not dedicated to simulating this specific mechanism, and the results do not indicate a high level of landslide hazard in these particular areas.

All the results show that future global change scenarios imply significant evolution in landslide hazards, regardless of the climate scenario or the period considered. Generally, the strongest effects on landslide hazards are linked to the evolution of

the climate rather than the evolution of land use; indeed, we have previously seen that, despite the incorporation of the effects of land use in the models, its impact remains quite limited, with an evolution of the FoS ranging from -0.1 to +0.1. These limited changes are, however, significant, depending on the future land use linked to the socioeconomic scenarios. In particular, the reduction in human presence (e.g., "Abandonment of the territory" scenario) results in an increase in slope stability. In contrast, the increase in anthropic activity implies some contrasting evolutions of the area, with some reduction in slope

stability. The scenario "Sheep and woods" illustrates this aspect, with the destabilization of areas due to the development of open forest. These scenarios, which represent four possible land use evolution trajectories, indicate contrasting spatial evolution patterns.

In contrast, apart from the impacts of land use change on landslide hazards, the quantification of future changes caused by

climate change indicates more significant changes in the size of the area that is susceptible to landslides; the size of this area depends on the landslide typology considered. Compared to the current period, the size of the area that is prone to deep landslides is larger in the future than the area prone to small landslides (both rotational and translational). On the other hand, the increase rate of areas prone to landslides is higher for the small landslide typology than for the deep landslide typology; indeed, the area classified in the 3 highest hazard levels is projected to increase approximately 6-fold from 2010 to 2040

(RCP8.5) and 8-fold from 2010 to 2100 (RCP8.5) for small rotational landslides, whereas the area classified in the 3 highest hazard levels is projected to increase approximately 1.5-fold from 2010 to 2040 (RCP8.5) as well as to 2100 (RCP8.5) for deep rotational landslides. This result confirms the analyses of Mc Innes et al. (2007); Moore et al., 2007; Crozier (2010), which suggested that the future evolution of landslide hazards in any environment depends on the type of landslide, and in particular on its depth and volume.

Fig 12 reflects the mean tendency of the future scenarios within a 30-year period. Another feature to be analysed is the occurrence frequency landslide events. Indeed, as the hydrogeological model considers the antecedent water content, an increase in total rainfall implies that there will be an increase in the frequency of the highest water table level, as indicated in Fig 5. As a result, when the antecedent water content is high, less water is required to trigger movement (Crozier, 2010). Therefore, landslide will occur more frequently. Figs 13 and 14 correspond to the highest frequency of the water filling ratio




for each period (0.5 for the 1980-2010 period; 0.6 for the 2021 - 2050 period (scenario RCP 8.5); 0.7 for the 2071 - 2100 period (scenario RCP 8.5)). These results indicate that the highest frequency of landslide hazards corresponding to the 2021 - 2050 period (scenario RCP 8.5) (Fig 13 b and Fig 14 b) will increase 1.5-fold, whereas the highest frequency of landslide hazards corresponding to the 2071 - 2100 period (scenario RCP 8.5), (Fig 13 c) and Fig 14 c)) will increase 4-fold.

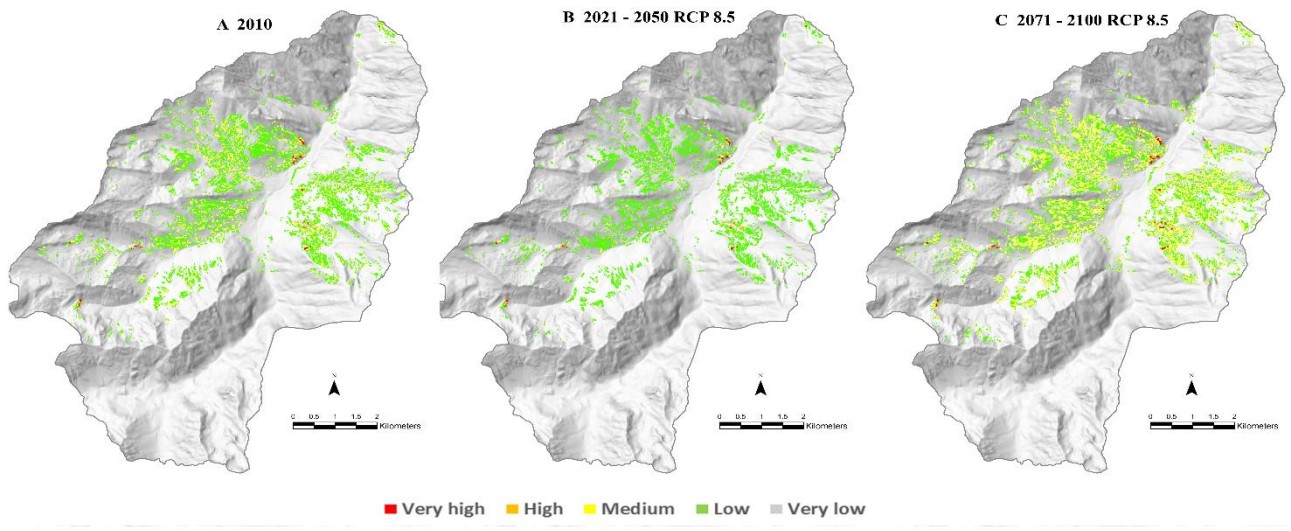

**Fig 13 – Landslide hazard levels corresponding to the highest frequency of the water filling ratio for each period – Shallow rotational landslide**

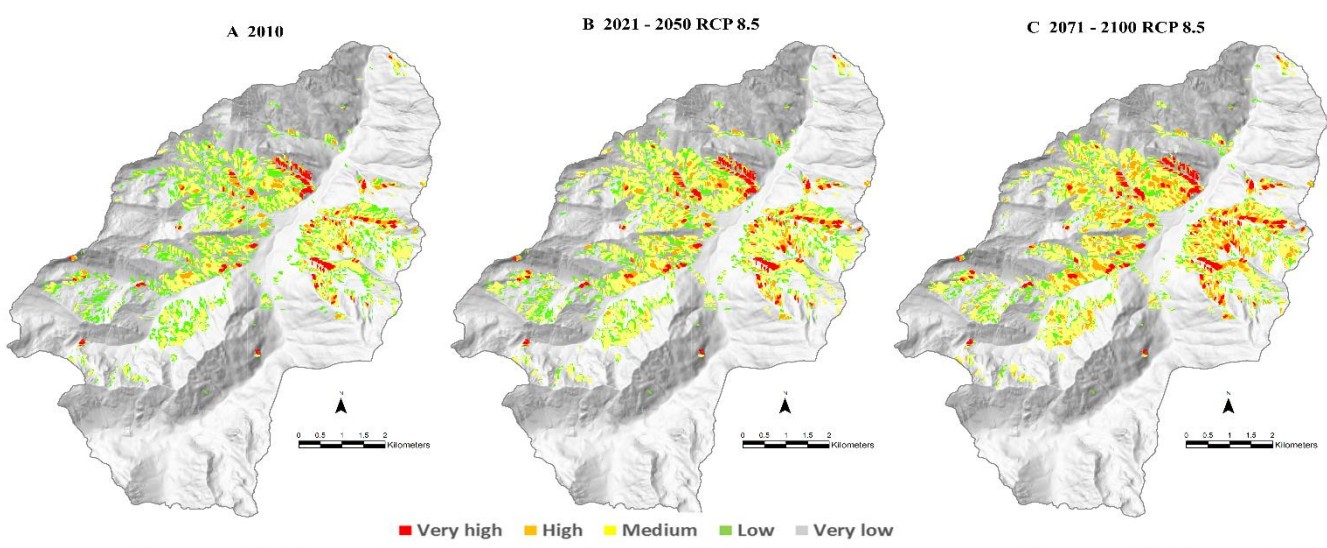

**Fig 14 – Landslide hazard levels corresponding to the highest frequency of the water filling ratio for each period – Deep rotational**
10
**landslide**





These significant results have to be analysed in consideration of the tested hypotheses. Some land use effects are considered in this study, but not all. The effects of the roots on slope stability is considered in the analysis, but it could be more accurate to consider the effects of vegetation succion linked with the runoff/infiltration balance, which is not spatially realized in this study. The infiltration capacity in forested areas may be higher than that in areas without forest; in some cases, this may result

in a reduction of slope stability, despite the soil reinforcement from roots (Crozier, 2010).

These results must also be put in perspective of the accuracy of the data. It must be noted that some uncertainties remain in this study with the lack of on-site hydrogeological knowledge to validate the hypotheses. Even if the hypotheses are not totally validated, we can assume that the comparisons between future and current states of hydrogeological context and the stability

of the slopes remain valid as we analyse trend evolution.

Some uncertainties are also associated with the future climatic data used, as the mountainous area has high spatial variability, with large changes in elevation over short distances. The 8 km resolution of the input data could be improved, and it may be of high interest to consider data with a higher resolution. Future results from the Climpy project (https://opcc-ctp.org/en/climpy) will surely reduce uncertainties in the input data. Moreover, a finer analysis of the balance of rainfall/snow

would greatly improve the accuracy of the results.

## 7   Conclusions

The present work provides a methodology for quantifying the impacts of global change on a mountainous region at the valley scale. It evaluates and quantifies the influences of both vegetation cover and climate on landslide activities projected to 2100. First, the results demonstrate the influence of land use on slope stability through the presence and type of forest. Future the

land use changes may lead to modifications in slope stability. Some increase in the stability could be observed in the areas where the forest is developing, whereas slope stability is decreasing in the areas where the forest is disappearing. These changes are significant, although they remain quite limited, as the evolution of the FoS ranges from -0.1 to +0.1.

Climate change may have a significant impacts on the water content of the soil; the results indicate a reduction in the FoS in a large part of the study area, depending on the landslide typology considered. Regardless of the scenario considered, there is a

decrease in slope stability; the drop in FoS evolution ranges from 0.1 to 0.2. Even in a case where the future evolution of the forest stabilizes the slopes, the evolution of the groundwater table destabilizes the area more. These changes are not uniform over the area and are particularly significant for the most extreme scenario, RCP 8.5.

Compared to the current period, the surface of the area that is prone to deep landslides is higher in the future than the area prone to small landslides (both rotational and translational). On the other hand, the increase rate of areas prone to landslides is

higher for small landslides than for deep landslides; indeed, the area classified in the three highest hazard levels is projected to increase approximately 6-fold from 2010 to 2040 (RCP8.5) and 8-fold from 2010 to 2100 (RCP8.5) for small rotational





landslides, whereas the area classified in the three highest hazard levels is projected to increase approximately 1.5-fold from 2010 to 2040 (RCP8.5) and to 2100 (RCP8.5) for the deep rotational landslides.

Interestingly, the evolution of extreme events is related to the occurrence frequency of the highest water filling ratio, which will increase. The results indicate that the highest frequency of landslide hazards for the 2021 - 2050 period (scenario RCP 8.5, Fig 13 b and Fig 14 b), will increase 1.5-fold, whereas the highest frequency of landslide hazards for the 2071 - 2100 period (scenario RCP 8.5, Fig 13 c and Fig 14 c), will increase 4-fold.

The uncertainties associated with the future climatic data used constitute an issue that should be addressed. Forthcoming results from the Climpy project will surely reduce uncertainties in the climate data. Moreover, a finer analysis of the rainfall/snow balance, based on a more accurate hydrogeological model, will greatly improve the accuracy of the results, including the seasonality analysis.

This study constitutes the first step in the process of risk assessment for different climate and economic development scenarios to evaluate the resilience of this region. All these methods have been implemented in a web-application platform (Grandjean et al., 2018) dedicated to stakeholders that will allow the stakeholders to obtain indicators on the best solutions for improving the resilience of valleys that are coping with global change.

## 8 Acknowledgements

This research was funded through the ANR (French Research Agency) project SAMCO ("Society Adaptation for coping with Mountain risks in a global change Context") n° ANR -12-SENV-0004-01. For more information, see http://www.anr-samco.com

The authors would like to thank the collaborators Hélène Bessière and Rodrigo Pedreros for their valuable help.

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
