# Peer review of "Modelling landslide hazard under global change: the case of a Pyrenean valley"

_Natural Hazards and Earth System Sciences, 2019_

## Referee Comment (RC1) · Anonymous Referee #1 · 28 Feb 2020

I've carefully read the manuscript "Modelling landslide hazard under global change: the case of a Pyrenean valley", by Séverine Bernardie and co-authors. I've found it interesting and I believe it might be interesting to all NHESS readers. It deals with a debated topic, i.e. the evaluation of changes in landslide hazard due to global change (both climatic and environmental), proposing a quantitative approach.

Overall, I've found the whole manuscript a bit lengthy. I believe that it could be strongly shortened, in particular, in the introduction (Section 1) and in the description of the study area (Section 3, e.g. lines 4-30 at page 7 are not very useful). Also the other sections could be slightly shortened. Moreover, the abstract is very long and should be reduced, at least reaching the 70% of the current length.

In my opinion, the whole article misses homogeneity. The numerous figures are very

heterogeneous: as an example, the scalebars in the maps are different among each other, as well as the dimensions of the figures themselves; in some maps, north arrow and scalebar are present and in some other they are missing; the used colors do not allow an immediate comprehension of the maps. I would suggest a review of all the figures (please see also all my comments below).

Moreover, several language issues are present. As an example, Authors use small and shallow as synonymous when referring to landslides, as well as big and deep. However, this is not correct: e.g. a shallow landslide is not always small. Furthermore, there is a bit of confusion in the manuscript among the terms hazard and susceptibility, which are not synonymous. Finally, also land use and land cover are not exactly synonymous. I would suggest using a rigorous terminology and correcting accordingly all the manuscript. Furthermore, I would suggest using "method" instead of "methodology" and "type" instead of "typology" everywhere in the text. Moreover, I would suggest using only "rainfall", uncountable. Finally, I would suggest to check all the acronyms used in the text.

Besides these comments regarding the structure and the terminology of the paper (even if the confusion between hazard and susceptibility is not only a matter of terminology but is a matter of method), I have some other comments regarding the proposed methods and procedures.

I have not understood why the Authors used two different periods for the short-term analysis: 2011-2040 for land use/cover change analysis and 2021-2050 for climate change analysis. I would suggest using the same period, also for a better comparison.

Regarding the ALICE model, it is not clear which parts of it are already present in the literature and which parts are introduced in the mentioned paper that is still under review. This should be better clarified.

Regarding the data, at the end of page 7 Authors state that 426 landslides were identified. However, the sum of the landslides reported in Table 1 is 346. Please clarify and

correct.

Furthermore, at line 12 (page 8) Authors state that only landslides with translational and rotational shear surfaces were selected to be modelled. However, at line 20, they state that four types are considered. Please clarify.

I would suggest to better clarify the procedure adopted for defining the maps of the factor of safety. It seems that some steps are missing.

In section 5, Authors state that the simulations were made for each landslide type and according to the 10 ground water filling ratios. Why? Why not considering only the most frequent ratios (as shown in fig.5)? Moreover, I see in the manuscript different analysis for the four groups of landslide types. It would be interesting to provide also a comprehensive analysis considering all the landslides together. I suggest producing also current and future maps showing the results obtained considering all the landslides together.

Still in Section 5, it is not clear how the five classes of hazard are defined. What does a "high" hazard mean in quantitative terms? Please explain.

Finally, I would suggest performing another analysis considering both climate and land use changes together, at least for the long-term period, so evaluating the combined effect of the future scenarios. If feasible, this would represent an element of innovation for the paper, and would improve its appeal.

Overall, I believe that the manuscript deserves publication after a thorough revision. Besides the above reported general comments, I made a list of some specific comments.

Page 1, line 22: please add "change" before the acronym LULC to be consistent.

Page 1, line 23: "climate change inputs" is somehow vague.

Page 1, line 27: "significant despite being small" is also vague.

Page 2, line 3: what do you mean with "small" landslides? This should be better explained.

Page 3, line 1: "Gariano and Guzzetti (2016)" is not reported in the reference list.

Page 3, line 9; please correct "rainfalls" here and elsewhere in the text.

Page 3, line 15: It seems to me that the 3 cited references do not deal with future evolution of hydro-meteorological conditions that implies modification in the frequency of landslide. I would suggest replacing them with the following examples: Alvioli et al., 2018 https://doi.org/10.1016/j.scitotenv.2018.02.315; Gariano et al., 2017 http://dx.doi.org/10.1016/j.scitotenv.2017.03.103; Peres and Cancelliere 2018, https://doi.org/10.1016/j.jhydrol.2018.10.036; Rianna et al. 2017, doi:10.3390/hydrology4030034; Robinson et al., 2017 dx.doi.org/10.1139/cgj-2015-0602, Sangelantoni et al. 2018, https://doi.org/10.1007/s11069-018-3328-6, Turkington et al., 2016 https://doi.org/10.1007/s10584-016-1657-6.

Page 3, line 18-24: I would suggest also reading the works by Gariano et al. 2018, https://doi.org/10.1007/s10113-017-1210-9; Persichillo et al. 2017, https://doi.org/10.1016/j.scitotenv.2016.09.125; Pisano et al. 2017, http://dx.doi.org/10.1016/j.scitotenv.2017.05.231; Promper et al. 2014, https://doi.org/10.1016/j.apgeog.2014.05.020, Reichenbach et al. 2014, https://doi.org/10.1007/s00267-014-0357-0.

Page 4, line 2: I would suggest repeating here some of the above-mentioned references on future impact of climate change on landslides, which are more recent that the cited ones.

Page 4, line 26: please replace "GIEC" with "IPCC".

Page 4, lines 29-32: I would suggest deleting the acronyms ALICE and GARDENIA here, since they are explained in the next paragraphs. Conversely, please add the meaning of the acronym FoS.

Page 6, line 16: intensity and occurrence in space? In time? Please explain.

Page 7, lines 4-30: please reduce this part.

Page 8, line 5: API is not defined.

Page 8, lines 16-17: this part should be deleted.

Page 13, lines 28-29: I would use always the same number of decimal digits, here and elsewhere in the text.

Page 14, line 14: the two selected RCP are not exactly contrasting, as reported. RCP4.5 is a "mid-way" scenario, while RCP8.5 is a "business as usual" scenario. If RCP2.6 and RCP8.5 were selected I would have said that they were contrasting.

Page 15, line 1: "increase in extreme precipitation events" is quite vague. I would specify if the increase is in the number, the frequency, or in something else.

Page 15, lines 7-10 (and figure 4): is this differentiation among solid and liquid precipitation useful?

Page 16, line 16: please note that you are mentioning Fig. 8 before Fig. 7. Please correct.

Page 17, line 1: I would add a reference to Fig. 6A after "translational typology" (that should be corrected into "translational type"). Analogously, I would refer to Fig.6B, 6C and 6D where the hazard of the other type is discussed after in the paragraph.

Page 17, line 20: only figure 7 show maps. Please correct.

Page 17, line 7: "the area that is susceptible to landslides".

Page 18, line 15-16: I would move the definition of TPR and FPR before the description of ROC curve.

Page 20, line 11: "Climate conditions are constant and set to the actual conditions". This sentence is not very clear.

Page 28, line 19-20: correct "Future the land use".

Page 28, line 28: I would replace "surface" with "percentage".

Tables

Table 2. If the table reports the "main predisposing factors of each landslide" – as the caption says – what's the meaning of the first row reporting all the information on the landslides themselves? I would suggest deleting this first row.

Table 3. I would replace "Classification according to Houet et al., 2017" with "Classification (Houet et al., 2017).

Figures

Figure 1. In the caption I would correct "landslide inventory" with "examples of inventoried landslides". Moreover, in panel E, I would suggest using non-overlapping values for the boundaries of the classes. Finally, check the font size to ensure readability.

Figure 2. I would add simple labels for the 4 scenarios, e.g. only Scenario 1, 2, 3, 4.

Figure 3. Also in this case I would use simple labels for the 4 scenarios, e.g. only Scenario 1, 2, 3, 4. Moreover, I would use only one decimal digit. Finally, in order to improve the readability of the graph, I would split it in two, let's say up and down, according to the two years 2040 and 2100. The dotted lines could be removed.

Figure 4. I would suggest adding labels for RCP4.5 and RCP8.5 in the respective columns and for short- and long-term in the respective rows. The readability of the figure will benefit from this. Moreover, I would suggest using non-overlapping values for the class boundaries. As an example, a point with a mean annual precipitation of 1000 mm is in the first or in the second class? Please correct.

Figure 5. In order to improve the readability of the figure I would suggest using similar colors for 2021-2050 and 2071-2100, respectively. As an example: two shades of red for 2021-2050 (lighter for RCP4.5 and darker for RCP8.5) and two shades of blue for

2071-2100 (lighter for RCP4.5 and darker for RCP8.5).

Figure 6. Please add labels for y-axes. The whole figure seems a bit compressed, please check. How the five hazard levels were defined?

Figure 7. Please note the caption of the figure says "hazard" while the legend says "susceptibility" and the values represent Factor of Safety. The two terms hazard and susceptibility are not synonymous! Please correct. Probably, "maps of FoS values" would be the best way to describe the map in its legend.

Figure 9. I would add two labels (2040 and 2100) in the two graph rows. Please note that the scalebar is not readable. I would suggest using a scalebar as in Figure 7, which is smaller (use only km instead of kilometers). Figures 10 and 11. Why not using the same symbology used in Figure 9 (only two classes, one for increase and another for decrease). I believe this would increase the immediate comprehension of the figures and the comparison among them. Alternatively, if the Authors want to maintain this difference, I would suggest using non-overlapping values for the class boundaries. As an example, a point with a FoS equal to -0.1 is in the first or in the second class? Please correct. Please note that the scalebar is not readable. I would suggest using a scalebar as in Figure 7, which is smaller (use only km instead of kilometers).

Figure 12. Please correct the x-axes labels ("pourcentage"). I would suggest using only one decimal digit. The legend seems a bit stretched and the green color in the legend seems different from the bars, please check.

Figures 13 and 14. Please note that the scalebar is not readable. I would suggest using a scalebar as in Figure 7, which is smaller (use only km instead of kilometers).

---

## Referee Comment (RC2) · Anonymous Referee #2 · 8 Apr 2020

The manuscript show a study of the influence of climate change on slope stability in a valley of the Pyrenees. Authors describe a very inspiring exercise integrating different inputs and models to simulate the effects on slopes' propensity to failure of possible future land use and precipitation scenarios. To do this, they use up-to-date tools based on spatially distributed models and perform a complete procedure to achieve their objectives. The article is a model for how such work could be conducted in other areas and is a suitable contribution for the journal. I have only several corrections and suggestions that I proceed to expose.

1. There is a major problem with the terminology throughout the manuscript. Authors use the term "landslide hazard" but they did not estimated that in their study stricto sensu. Hazard implies spatio-temporal probability. Authors are really estimating the

change in the Factor of Safety (FoS) (i.e. Slope Stability) of the slopes according to different conditions. It is true that the results of their calculations are spatially distributed and they are providing temporal information. Nevertheless, their model outputs are not the expected number of landslides per year and per area. The nature of the data have its implications because, for example, the FoS do not serve to estimate risk. If authors want to be precise, they have to use in the text and in the title the term "slope stability" instead of "landslide hazard".

2. Many researchers have described how anthropic activities have high impact on the stability of slopes (cf. Glade, 2003; Remondo et al., 2005). Crozier (2010) state "Changes resulting from human activity are seen as a factor of equal, if not greater, importance than climate change in affecting the temporal and spatial occurrence of landslides". This is reasonable because slope modifications due to infrastructure construction or urbanization and significant land use changes produce great alteration on slope conditions. Please, discuss your results taking this paradigm in your mind. In the presented study area the human activities have a minimal disturbance to the environment, which may explain that the increase of precipitation due to climate change could have more impact than human action. This is not the situation in many countries, specially across the Global South. This idea must be stressed because, if not, other researchers can underestimate the human action over the physical medium.

3. Authors explain in the introduction section that there are two ways to simulate future scenarios of landslide activity: physical and statistical models. They use an approach based on physical modelling to investigate failure processes at regional scale. I suggest authors to justify the selection of a physical model and discuss about other approaches. To do so, I suggest them to consult several papers about comparisons between physical and statistical models (e.g. Cervi et al., 2010; Zizioli et al., 2013; Davis and Blesius, 2015; Ciurleo et al., 2017; Bartelleti et al., 2017; Galve et al., 2017; Oliveira et al., 2017).

4. In order to enrich the literature and discussion of the manuscript, I suggest authors

to read the following papers dealing with the effects of land use change on landslide susceptibility and hazard: Vanacker et al., 2003; Van Beek and Van Ash, 2004; Reichenbach et al., 2014; Galve et al., 2015; Persichillo et al, 2017.

5. I also suggest authors to discuss about the application of their model and the extrapolation of their results to other regions (data and model requirements).

6. It is needed a large map where all the cited toponyms are included.

7. I do not like how authors describe landslide typology and morphologies. For example, they use "landslides with rotational shear surfaces, landslides with translational shear surfaces". Why are they using this long descriptions if they can use widely accepted landslide classifications such as Cruden & Varnes (1996) or Hungr et al. (2014)? Regarding the landslide associated landforms they use "(i) the landslide-triggering zone (LTZ) and (ii) the landslide accumulation zone (LAZ)" to designate parts of the mapped landslides. However the term accepted by the international community for their "LTZ" and "LAZ" should be "Zone of depletion" and "Zone of accumulation" (Varnes, 1978). The use of appropriate and widely accepted terminology avoid the necessity of explaining the not so widely used terms, as authors have to do in the second paragraph of page 8.

8. Models seem to indicate that "Bare soils" are always stable. Please, explain that?

9. I would appreciate a table with the model validation results and a figure with the ROC and PRC curves. How can explain the high performance of the models? In my opinion, the prediction capability is very good for a physical model applied at regional scale.

—

OTHER COMMENTS

Table 2 "Defined using related literature based on field investigations". Local or global literature?

Figure 1. - Colour landslide according to their type. - Add coordinates. - Authors only mapped active landslides? - Caption: Change "layer" and "layers" by "deposits".

Figure 2. - Change "Mineral surfaces" by "Bare Rock" (as Corine Land Cover terminology)

Table 3 How was additional cohesion calculated?

Section 3 "Gave" is a term used for creeks or streams in the western Pyrenees. Please, change the term to the appropriate English word or define "Gave" in the text. Please, define what the "Soum de Grum" and the "Grand Barbat" are. Are they a place, an area, a district, a landform?

Page 4 Line 25. What is GIEC? Lines 29 and 31. Assign citations to ALICE and GARDENIA tools.

Page 6 Line 21. What are the RTM services?

Page 9 Line 10. Please, define "moraine colluviums"?

Page 14 Line 15. Add a citation to ALADIN-Climate model of Météo-France.

Page 16 Line 14. Please, explain the method applied to define the hazard classes.

Page 17 Reducing the first paragraph could make the reading more fluent.

Page 18 Lines 3-16 aprox. This is an explanation of the validation techniques and it may be displaced to the metholodology section. In this regard, how were no-landslide/stable points selected to produce ROC curves?

REFERENCES

Bartelletti, C., Galve, J.P., Barsanti, M., Giannecchini, R., Avanzi, G.D.A., Galanti, Y., Cevasco, A., Azañón, J.M. and Mateos, R.M., 2017, May. GIS-Based Deterministic and Statistical Modelling of Rainfall-Induced Landslides: A Comparative Study. In Workshop on World Landslide Forum (pp. 749-757). Springer, Cham.

Cervi, F., Berti, M., Borgatti, L., Ronchetti, F., Manenti, F. and Corsini, A., 2010. Comparing predictive capability of statistical and deterministic methods for landslide susceptibility mapping: a case study in the northern Apennines (Reggio Emilia Province, Italy). Landslides, 7(4), pp.433-444.

Ciurleo, M., Cascini, L. and Calvello, M., 2017. A comparison of statistical and deterministic methods for shallow landslide susceptibility zoning in clayey soils. Engineering Geology, 223, pp.71-81.

Crozier, M.J., 2010. Deciphering the effect of climate change on landslide activity: A review. Geomorphology, 124(3-4), pp.260-267.

Cruden, D.M. and Varnes, D.J., 1996. Landslides: investigation and mitigation. Chapter 3-Landslide types and processes. Transportation research board special report, (247).

Davis, J. and Blesius, L., 2015. A hybrid physical and maximum-entropy landslide susceptibility model. Entropy, 17(6), pp.4271-4292.

Galve, J.P., Bartelletti, C., Notti, D., Fernández-Chacón, F., Barsanti, M., Azañón, J.M., Pérez-Peña, V., Giannecchini, R., Avanzi, G.D.A., Galanti, Y. and Lamas, F.J., 2017, May. Deterministic and Probabilistic Slope Stability Models Forecast Performance at~ 1: 5000-Scale. In Workshop on World Landslide Forum (pp. 741-748). Springer, Cham.

Galve, J.P., Cevasco, A., Brandolini, P. and Soldati, M., 2015. Assessment of shallow landslide risk mitigation measures based on land use planning through probabilistic modelling. Landslides, 12(1), pp.101-114.

Glade, T., 2003. Landslide occurrence as a response to land use change: a review of evidence from New Zealand. Catena, 51(3-4), pp.297-314.

Oliveira, S., Zêzere, J., Lajas, S. and Melo, R., 2017. Combination of statistical and physically based methods to assess shallow slide susceptibility at the basin scale.

Natural Hazards and Earth System Sciences, 17(7), pp.1091-1109.

Reichenbach, P., Mondini, A.C. and Rossi, M., 2014. The influence of land use change on landslide susceptibility zonation: the Briga catchment test site (Messina, Italy). Environmental management, 54(6), pp.1372-1384.

Remondo, J., Soto, J., González-Díez, A., de Terán, J.R.D. and Cendrero, A., 2005. Human impact on geomorphic processes and hazards in mountain areas in northern Spain. Geomorphology, 66(1-4), pp.69-84.

Van Beek, L.P.H. and Van Asch, T.W., 2004. Regional assessment of the effects of land-use change on landslide hazard by means of physically based modelling. Natural Hazards, 31(1), pp.289-304.

Vanacker, V., Vanderschaeghe, M., Govers, G., Willems, E., Poesen, J., Deckers, J. and De Bievre, B., 2003. Linking hydrological, infinite slope stability and land-use change models through GIS for assessing the impact of deforestation on slope stability in high Andean watersheds. Geomorphology, 52(3-4), pp.299-315.

Varnes, D.J. 1978. Slope Movement Types and Processes. In Special Report 176: LAndslides: Analysisand Control (R.L. Schuster and R.J. Krizek, eds.), TRB, National Research Council, Washington, D.C., pp. 11-33.

Zizioli, D., Meisina, C., Valentino, R. and Montrasio, L., 2013. Comparison between different approaches to modeling shallow landslide susceptibility: a case history in Oltrepo Pavese, Northern Italy. Natural Hazards and Earth System Sciences, 13(3), p.559.

---

## Author Comment (AC1) · 11 Jul 2020

Dear Referee #1,

Please find below and in attached file our answers to each question and comment you have provided for the review of the paper. I thank you for your relevant and interesting comments.

with best regards,

Séverine Bernardie

Anonymous Referee #1

1. I've carefully read the manuscript "Modelling landslide hazard under global change:

[Figure]

the case of a Pyrenean valley", by Séverine Bernardie and co-authors. I've found it interesting and I believe it might be interesting to all NHESS readers. It deals with a debated topic, i.e. the evaluation of changes in landslide hazard due to global change (both climatic and environmental), proposing a quantitative approach.

Overall, I've found the whole manuscript a bit lengthy. I believe that it could be strongly shortened, in particular, in the introduction (Section 1) and in the description of the study area (Section 3, e.g. lines 4-30 at page 7 are not very useful). Also the other sections could be slightly shortened. Moreover, the abstract is very long and should be reduced, at least reaching the 70% of the current length.

The introduction will be shortened, but other additional elements may be introduced, in relation to Anonymous Referee #2 ; the description of the study area will be strongly shortened, as well as the abstract.

2. In my opinion, the whole article misses homogeneity. The numerous figures are very heterogeneous: as an example, the scalebars in the maps are different among each other, as well as the dimensions of the figures themselves; in some maps, north arrow and scalebar are present and in some other they are missing; the used colors do not allow an immediate comprehension of the maps. I would suggest a review of all the figures (please see also all my comments below).

All the figures have been revised according to this relevant comment and the comments concerning some specifically figures.

3. Moreover, several language issues are present. As an example, Authors use small and shallow as synonymous when referring to landslides, as well as big and deep. However, this is not correct: e.g. a shallow landslide is not always small. Furthermore, there is a bit of confusion in the manuscript among the terms hazard and susceptibility, which are not synonymous. Finally, also land use and land cover are not exactly synonymous. I would suggest using a rigorous terminology and correcting accordingly all the manuscript. Furthermore, I would suggest using "method" instead of "methodology" and "type" instead of "typology" everywhere in the text. Moreover, I would suggest using only "rainfall", uncountable. Finally, I would suggest to check all the acronyms used in the text.

The paper will be modified with considering harmonised adequate terminology ; the terms land cover, shallow, moderately deep and deep landslide, hazard, method, type, and rainfall have been kept.

4. Besides these comments regarding the structure and the terminology of the paper (even if the confusion between hazard and susceptibility is not only a matter of terminology but is a matter of method), I have some other comments regarding the proposed methods and procedures.

Concerning the terminology hazard/susceptibility, the paragraph Page 6 lines 14-16 is written to clarify and justify the choice of hazard ; to address this comment as well as the further comment on this subject, we will add some other elements, as indicated below : Landslide hazard assessment considers run-out, magnitude, and return period for a given intensity (Varnes, 1984). As in many cases, the hazard analysis is not completed. Notably, run-out is not accounted for in this study. Nevertheless, the landslide susceptibility assessment is converted into landslide hazard assessment by expert knowledge (Van Western et al., 2006, 2008, Corominas et al. 2014).

5. I have not understood why the Authors used two different periods for the short-term analysis: 2011-2040 for land use/cover change analysis and 2021-2050 for climate change analysis. I would suggest using the same period, also for a better comparison.

Among the few land use/land cover maps obtained, we have considered that those of 2040 and 2100 are the most representative of the period 2021-2050 and 2071-2100. It will be clarified in the paper in page 12 lines 10 - 11

6. Regarding the ALICE model, it is not clear which parts of it are already present in the literature and which parts are introduced in the mentioned paper that is still under

review. This should be better clarified.

In the paper Vandromme et al., 2020, one of the principal subject introduced is the development of the strategy, for improving the accuracy of the calibration and the validation. It will be added in the paper Page 5, lines 24. Moreover, this paper is now published ( https://doi.org/10.1016/j.geomorph.2020.107307 )

7. Regarding the data, at the end of page 7 Authors state that 426 landslides were identified. However, the sum of the landslides reported in Table 1 is 346. Please clarify and correct.

There were indeed a mistake : The majority of these landslides are considered active or may be triggered by rainfall (346 out of 426), the others being associated to more complex landslides, and not considered active any more. It will be clarified in the paper.

8. Furthermore, at line 12 (page 8) Authors state that only landslides with translational and rotational shear surfaces were selected to be modelled. However, at line 20, they state that four types are considered. Please clarify.

In the text it is written that four types are considered: i) shallow translational landslides, ii) shallow rotational landslides, iii) moderately deep and iv) deep landslides For better understanding we will add some terms in the text

9. In section 5, Authors state that the simulations were made for each landslide type and according to the 10 ground water filling ratios. Why? Why not considering only the most frequent ratios (as shown in fig.5)?

I completely agree with you that it is not explained why we provide the simulations for 10 ground water filling ratio ; the objective of this Figure is to demonstrate the sensitivity of the model to the evolution of the water filling ratio ; it shows that the evolution is different according to the 4 types of landslide ; moreover, we can see that the evolution is not linear, and that a threshold of behaviour appears in the results, depending on the type of landslide. Some explanations will be added.

10. Moreover, I see in the manuscript different analysis for the four groups of landslide types. It would be interesting to provide also a comprehensive analysis considering all the landslides together. I suggest producing also current and future maps showing the results obtained considering all the landslides together.

We will provide additional maps, considering all landslides, in figure 7 and in figure 15.

11. Still in Section 5, it is not clear how the five classes of hazard are defined. What does a "high" hazard mean in quantitative terms? Please explain.

Additional information on the definition of the classes of hazard will be incorporated (references), in particular this following table :

Landslide hazard class Value of simulation expressed in FoS Very high FoS ≤ 0.9 High 0.9 < FoS ≤ 1.1 Moderate 1.1 < FoS ≤ 1.35 Low 1.35 < FoS ≤ 1.5 Null FoS > 1.5

12. Finally, I would suggest performing another analysis considering both climate and land use changes together, at least for the long-term period, so evaluating the combined effect of the future scenarios. If feasible, this would represent an element of innovation for the paper, and would improve its appeal.

In section 5.2.2., the analysis of the impact of future climate on landslide hazard incorporate in fact the effect of land cover as well, since the scenario "abandonment of the area" is included in this analysis. Indeed the land cover map in 2040 is considered for the period analysis 2021-2050 ; the land cover map in 2100 is considered for the period analysis 2071-2100 ; this is not clearly explained, so we will modify the text accordingly. The results show that the effect of CC is more significant than the land cover ; that is why in the paper we have only presented the results with the combination of CC and 1 land cover scenario (scenario "abandonment of the area"), even if we have computed all the combinations.

Overall, I believe that the manuscript deserves publication after a thorough revision.

13. Besides the above reported general comments, I made a list of some specific

comments.

Page 1, line 22: please add "change" before the acronym LULC to be consistent.

Done

Page 1, line 23: "climate change inputs" is somehow vague.

Done and clarified – the abstract is anyway shortened and modified. Page 1, line 27: "significant despite being small" is also vague. Done and clarified – the abstract is anyway shortened and modified. Page 2, line 3: what do you mean with "small" landslides? This should be better explained.

It has been changed to "shallow"

Page 3, line 1: "Gariano and Guzzetti (2016)" is not reported in the reference list.

Done

Page 3, line 9; please correct "rainfalls" here and elsewhere in the text.

The "s" has been removed

Page 3, line 15: It seems to me that the 3 cited references do not deal with future evolution of hydro-meteorological conditions that implies modification in the frequency of landslide. I would suggest replacing them with the following examples: Alvioli et al., 2018 https://doi.org/10.1016/j.scitotenv.2018.02.315; Gariano et al., 2017 http://dx.doi.org/10.1016/j.scitotenv.2017.03.103; Peres and Cancelliere 2018, https://doi.org/10.1016/j.jhydrol.2018.10.036; Rianna et al. 2017, doi:10.3390/hydrology4030034; Robinson et al., 2017 dx.doi.org/10.1139/cgj-2015-0602, Sangelantoni et al. 2018, https://doi.org/10.1007/s11069-018-3328-6, Turkington et al., 2016 https://doi.org/10.1007/s10584-016-1657-6.

Done – I thank you for these relevant references

Page 3, line 18-24: I would suggest also reading the works by Gariano et al. 2018, https://doi.org/10.1007/s10113-017-1210-9; Persichillo et al. 2017, https://doi.org/10.1016/j.scitotenv.2016.09.125; Pisano et al. 2017, http://dx.doi.org/10.1016/j.scitotenv.2017.05.231; Promper et al. 2014, https://doi.org/10.1016/j.apgeog.2014.05.020, Reichenbach et al. 2014, https://doi.org/10.1007/s00267-014-0357-0.

Done – I thank you for these relevant references

Page 4, line 2: I would suggest repeating here some of the above-mentioned references on future impact of climate change on landslides, which are more recent that the cited ones.

Done

Page 4, line 26: please replace "GIEC" with "IPCC".

Done

Page 4, lines 29-32: I would suggest deleting the acronyms ALICE and GARDENIA here, since they are explained in the next paragraphs. Conversely, please add the meaning of the acronym FoS. Done Page 6, line 16: intensity and occurrence in space? In time? Please explain.

Some explanations will be add in this part, to explain what is analysed in this study, and the choice of adequate term "hazard".

Page 7, lines 4-30: please reduce this part.

This part will be entirely removed

Page 8, line 5: API is not defined.

Done

Page 8, lines 16-17: this part should be deleted.

Done

Page 13, lines 28-29: I would use always the same number of decimal digits, here and elsewhere in the text.

Done, one decimal digit is considered

Page 14, line 14: the two selected RCP are not exactly contrasting, as reported. RCP4.5 is a "mid-way" scenario, while RCP8.5 is a "business as usual" scenario. If RCP2.6 and RCP8.5 were selected I would have said that they were contrasting.

Done – Yes we agree ; "contrasting" will be removed

Page 15, line 1: "increase in extreme precipitation events" is quite vague. I would specify if the increase is in the number, the frequency, or in something else.

Done

Page 15, lines 7-10 (and figure 4): is this differentiation among solid and liquid precipitation useful?

This analysis considering both solid and liquid precipitation has been done, since this information is considered and analysed within the hydrogeological model Gardenia.

Page 16, line 16: please note that you are mentioning Fig. 8 before Fig. 7. Please correct.

Done

Page 17, line 1: I would add a reference to Fig. 6A after "translational typology" (that should be corrected into "translational type"). Analogously, I would refer to Fig.6B, 6C and 6D where the hazard of the other type is discussed after in the paragraph.

Done

Page 17, line 20: only figure 7 show maps. Please correct.

Done

Page 17, line 7: "the area that is susceptible to landslides".

Done

Page 18, line 15-16: I would move the definition of TPR and FPR before the description of ROC curve.

Done

Page 20, line 11: "Climate conditions are constant and set to the actual conditions". This sentence is not very clear.

Reformulation done : "Climate conditions are considered equal between current and the 2 future periods

Page 28, line 19-20: correct "Future the land use".

Done

Page 28, line 28: I would replace "surface" with "percentage".

Done

Tables Table 2. If the table reports the "main predisposing factors of each landslide" – as the caption says – what's the meaning of the first row reporting all the information on the landslides themselves? I would suggest deleting this first row.

Done

Table 3. I would replace "Classification according to Houet et al., 2017" with "Classification (Houet et al., 2017).

Done

Figures Figure 1. In the caption I would correct "landslide inventory" with "examples of inventoried landslides". Moreover, in panel E, I would suggest using non-overlapping values for the boundaries of the classes. Finally, check the font size to ensure readability.

Done, except that we left "landslide inventory" as it constitutes the entire inventory of the area.

Figure 2. I would add simple labels for the 4 scenarios, e.g. only Scenario 1, 2, 3, 4.

Done

Figure 3. Also in this case I would use simple labels for the 4 scenarios, e.g. only Scenario 1, 2, 3, 4. Moreover, I would use only one decimal digit. Finally, in order to improve the readability of the graph, I would split it in two, let's say up and down, according to the two years 2040 and 2100. The dotted lines could be removed.

Done

Figure 4. I would suggest adding labels for RCP4.5 and RCP8.5 in the respective columns and for short- and long-term in the respective rows. The readability of the figure will benefit from this. Moreover, I would suggest using non-overlapping values for the class boundaries. As an example, a point with a mean annual precipitation of 1000 mm is in the first or in the second class? Please correct.

Done

Figure 5. In order to improve the readability of the figure I would suggest using similar colors for 2021-2050 and 2071-2100, respectively. As an example: two shades of red for 2021-2050 (lighter for RCP4.5 and darker for RCP8.5) and two shades of blue for 2071-2100 (lighter for RCP4.5 and darker for RCP8.5). Done Figure 6. Please add labels for y-axes. The whole figure seems a bit compressed, please check. How the five hazard levels were defined?

Done – explanations of the 5 hazard levels will be indicated in the text.

Figure 7. Please note the caption of the figure says "hazard" while the legend says "susceptibility" and the values represent Factor of Safety. The two terms hazard and

susceptibility are not synonymous! Please correct. Probably, "maps of FoS values" would be the best way to describe the map in its legend.

We removed "susceptibility" term, and left the term "hazard"; the explanations of this choice is indicated in Page 6.

Figure 9. I would add two labels (2040 and 2100) in the two graph rows. Please note that the scalebar is not readable. I would suggest using a scalebar as in Figure 7, which is smaller (use only km instead of kilometers).

Done

Figures 10 and 11. Why not using the same symbology used in Figure 9 (only two classes, one for increase and another for decrease). I believe this would increase the immediate comprehension of the figures and the comparison among them. Alternatively, if the Authors want to maintain this difference, I would suggest using non-overlapping values for the class boundaries. As an example, a point with a FoS equal to -0.1 is in the first or in the second class? Please correct. Please note that the scalebar is not readable. I would suggest using a scalebar as in Figure 7, which is smaller (use only km instead of kilometers).

Done – we maintain the 4 classes, as the difference between maps are more important (until 0.2) than in figure 9 (until 0.1)

Figure 12. Please correct the x-axes labels ("pourcentage"). I would suggest using only one decimal digit. The legend seems a bit stretched and the green color in the legend seems different from the bars, please check.

Done

Figures 13 and 14. Please note that the scalebar is not readable. I would suggest using a scalebar as in Figure 7, which is smaller (use only km instead of kilometers). Done
[Figure]

Please also note the supplement to this comment:
https://www.nat-hazards-earth-syst-sci-discuss.net/nhess-2019-311/nhess-2019-311-AC1-supplement.pdf

---

## Author Comment (AC2) · 11 Jul 2020

Dear Referee #2,

Please find below and in attached file our answers to each question and comment you have provided for the review of the paper. I thank you for your relevant and constructive comments.

with best regards,

Séverine Bernardie

Anonymous Referee #2

The manuscript show a study of the influence of climate change on slope stability in a

valley of the Pyrenees. Authors describe a very inspiring exercise integrating different inputs and models to simulate the effects on slopes' propensity to failure of possible future land use and precipitation scenarios. To do this, they use up-to-date tools based on spatially distributed models and perform a complete procedure to achieve their objectives. The article is a model for how such work could be conducted in other areas and is a suitable contribution for the journal. I have only several corrections and suggestions that I proceed to expose.

1. There is a major problem with the terminology throughout the manuscript. Authors use the term "landslide hazard" but they did not estimated that in their study stricto sensu. Hazard implies spatio-temporal probability. Authors are really estimating the change in the Factor of Safety (FoS) (i.e. Slope Stability) of the slopes according to different conditions. It is true that the results of their calculations are spatially distributed and they are providing temporal information. Nevertheless, their model outputs are not the expected number of landslides per year and per area. The nature of the data have its implications because, for example, the FoS do not serve to estimate risk. If authors want to be precise, they have to use in the text and in the title the term "slope stability" instead of "landslide hazard".

Concerning the terminology hazard, the paragraph Page 6 lines 14-16 was written to clarify and justify the choice of hazard ; to address this comment as well as the further comment on this subject, we will had some other elements, as indicated below : Landslide hazard assessment considers run-out, magnitude, and return period for a given intensity (Varnes, 1984). As in many cases, the hazard analysis is not completed. Notably, run-out is not accounted for in this study. Nevertheless, the landslide susceptibility assessment is converted into landslide hazard assessment by expert knowledge (Van Westen et al., 2006, 2008, Corominas et al. 2014)

2. Many researchers have described how anthropic activities have high impact on the stability of slopes (cf. Glade, 2003; Remondo et al., 2005). Crozier (2010) state "Changes resulting from human activity are seen as a factor of equal, if not greater,

importance than climate change in affecting the temporal and spatial occurrence of landslides". This is reasonable because slope modifications due to infrastructure construction or urbanization and significant land use changes produce great alteration on slope conditions. Please, discuss your results taking this paradigm in your mind. In the presented study area the human activities have a minimal disturbance to the environment, which may explain that the increase of precipitation due to climate change could have more impact than human action. This is not the situation in many countries, specially across the Global South. This idea must be stressed because, if not, other researchers can underestimate the human action over the physical medium.

On this topic we have discussed about the effects that are not considered in this approach (vegetation succion with runoff/infiltration balance). But I completely agree on your comment, that the modification of the land cover is quite small on this territory (as seen in figure 3). Moreover, some anthropic modification that may appear, such as slope modifications, are indeed not considered in this approach ; this will be stressed and argued in the discussion.

3. Authors explain in the introduction section that there are two ways to simulate future scenarios of landslide activity: physical and statistical models. They use an approach based on physical modelling to investigate failure processes at regional scale. I suggest authors to justify the selection of a physical model and discuss about other approaches. To do so, I suggest them to consult several papers about comparisons between physical and statistical models (e.g. Cervi et al., 2010; Zizioli et al., 2013; Davis and Blesius, 2015; Ciurleo et al., 2017; Bartelletti et al., 2017; Galve et al., 2017; Oliveira et al., 2017).

I thank you for these interesting papers, that will be added in this part of the paper ; we have indeed focused the discussion on the fact that physical models permit to quantify the impact of future climate and socio-economical scenario into landslide hazard ; but we will complete this part with integrating justification of the choice of physical models by providing some information on the performance of the models and the assumptions

made.

4. In order to enrich the literature and discussion of the manuscript, I suggest authors to read the following papers dealing with the effects of land use change on landslide susceptibility and hazard: Vanacker et al., 2003; Van Beek and Van Ash, 2004; Reichenbach et al., 2014; Galve et al., 2015; Persichillo et al, 2017.

I thank you for these relevant papers. They will be included within the introduction section.

5. I also suggest authors to discuss about the application of their model and the extrapolation of their results to other regions (data and model requirements).

I completely agree with this comment ; this part is missing from the paper and will be added.

6. It is needed a large map where all the cited toponyms are included.

Finally, as the paragraph which concerns the description of the site has been strongly shortened, all cited toponyms have been deleted. Thus the map is not necessary now.

7. I do not like how authors describe landslide typology and morphologies. For example, they use "landslides with rotational shear surfaces, landslides with translational shear surfaces". Why are they using this long descriptions if they can use widely accepted landslide classifications such as Cruden & Varnes (1996) or Hungr et al. (2014)? Regarding the landslide associated landforms they use "(i) the landslide triggering zone (LTZ) and (ii) the landslide accumulation zone (LAZ)" to designate parts of the mapped landslides. However the term accepted by the international community for their "LTZ" and "LAZ" should be "Zone of depletion" and "Zone of accumulation" (Varnes, 1978). The use of appropriate and widely accepted terminology avoid the necessity of explaining the not so widely used terms, as authors have to do in the second paragraph of page 8.

We will indeed refer to Cruden & Varnes classification and reduce the description.

The landslide triggering zone (LTZ) and the landslide accumulation zone (LAZ) will be replaced by "Zone of depletion" and "Zone of accumulation" (Varnes, 1978).

8. Models seem to indicate that "Bare soils" are always stable. Please, explain that?

I agree, this is a mistake : this is "Bare rock", and not 'Bare soil' ; it has been changed in Page line.

9. I would appreciate a table with the model validation results and a figure with the ROC and PRC curves. How can explain the high performance of the models? In my opinion, the prediction capability is very good for a physical model applied at regional scale.

AUC has been calculated, as well as other indicators according to Brenning, 2005, and provided in figure 8 ; it permits by that way to validate the quality of the model. We consider that these indicators permits to quantify the performance of the model.

— OTHER COMMENTS

Table 2 "Defined using related literature based on field investigations". Local or global literature? It concerns local literature ; it will be added in the table Figure 1. - Colour landslide according to their type. - Add coordinates. - Authors only mapped active landslides? - Caption: Change "layer" and "layers" by "deposits".

This figure has been modified accordingly. Colouring the landslide according to their type was not really visual in this figure. But in figure 7, all the landslides are separated between the 4 types.

Figure 2. - Change "Mineral surfaces" by "Bare Rock" (as Corine Land Cover terminology)

Done

Table 3 How was additional cohesion calculated?

The additional cohesion is determined from literature (Wu et al., 2004 ; Norris et al., 2008). It will be added in the paper.

Section 3 "Gave" is a term used for creeks or streams in the western Pyrenees. Please, change the term to the appropriate English word or define "Gave" in the text. Please, define what the "Soum de Grum" and the "Grand Barbat" are. Are they a place, an area, a district, a landform?

This section has been shortened as asked by the first referee. Thus all these terms have been removed.

Page 4 Line 25. What is GIEC?

It has been replaced by IPCC.

Lines 29 and 31. Assign citations to ALICE and GARDENIA tools.

Done

Page 6 Line 21. What are the RTM services?

RTM means Restauration des Terrains de Montagne; it is the French survey of hazard and forest management in mountainous territories, and constitutes a part from the French Forest Office (ONF). It will be added in the paper.

Page 9 Line 10. Please, define "moraine colluviums"?

There is a mistake in the text : it will be replaced by "moraine deposits or colluviums"

Page 14 Line 15. Add a citation to ALADIN-Climate model of Météo-France.

Done

Page 16 Line 14. Please, explain the method applied to define the hazard classes.

Additional information on the definition of the classes of hazard will be incorporated (references), in particular this following table :

Landslide hazard class Value of simulation expressed in FoS Very high FoS ≤ 0.9 High 0.9 < FoS ≤ 1.1 Moderate 1.1 < FoS ≤ 1.35 Low 1.35 < FoS ≤ 1.5 Null FoS > 1.5

Page 17 Reducing the first paragraph could make the reading more fluent.

Done

Page 18 Lines 3-16 aprox. This is an explanation of the validation techniques and it may be displaced to the metholodology section.

Done

In this regard, how were no-landslide/stable points selected to produce ROC curves?

In this approach, we consider the class very high and high hazard that can be compared to landslide inventory ; it corresponds to FoS < 1.1. It will be added in the paper.

Please also note the supplement to this comment:
https://www.nat-hazards-earth-syst-sci-discuss.net/nhess-2019-311/nhess-2019-311-AC2-supplement.pdf

---

## Author Response (AR2)

**Anonymous in acknowledgements of published article:** Yes No

**Recommendation to the editor**

**1) Scientific significance**
Does the manuscript represent a substantial contribution to the understanding of natural hazards and their consequences (new concepts, ideas, methods, or data)?

Excellent Good Fair Poor

**2) Scientific quality**
Are the scientific and/or technical approaches and the applied methods valid? Are the results discussed in an appropriate and balanced way (clarity of concepts and discussion, consideration of related work, including appropriate references)?

Excellent Good Fair Poor

**3) Presentation quality**
Are the scientific data, results and conclusions presented in a clear, concise, and well-structured way (number and quality of figures/tables, appropriate use of technical and English language, simplicity of the language)?

Excellent Good Fair Poor

For final publication, the manuscript should be
**accepted as is**.
accepted subject to **technical corrections**.
accepted subject to **minor revisions**.
reconsidered after **major revisions**:
    I am willing to review the revised paper.
    I am **not** willing to review the revised paper.
**rejected**.

**Suggestions for revision or reasons for rejection (will be published if the paper is accepted for final publication)**
Dear Authors,
I've carefully read the revised version of your manuscript.
Overall, the paper has been improved during the review phase. It gained homogeneity and clarity. All comments and suggestions made by both reviewers were addressed. There are a few last correction suggestions listed below.
I'm still doubtful about using the Factor of Safety directly as an index for landslide hazard. I see that in the mentioned references, in some cases of deterministic analysis, is suggested to determine landslide hazard using slope stability models, resulting in the calculation of factors of safety.
Therefore, I accept the answer proposed by the Authors and I leave the decision to the Editor.

List of minor corrections:
Title: I would say "changes"

*Down*
Page 1, line 18: I would say "emission scenarios"
*Down*
Page 1, line 22: I would say "statistically significant"
*Down*
Page 1, line 28: I would say "increasing rate"
*Down*
Page 2, line 4: I would say "changes"
*Down*
Page 2, line 10: I would say "Climate change affected and will affect…"
*Down*
Page 2, line 13: a point is missing
*Down*
Page 2, line 14: I would say "types" instead of "typologies" here and elsewhere in the text.
*Down in elsewhere in the text*
Page 3, line2 1-2: I would sort the references chronologically, as in the other cases.
*Down*
Page 4, line 27: please define "FoS" since it is the first time it is mentioned in the text (do not consider the abstract)
*Down*
Page 5, line 1: here you can use only "Fos"
*Down*
Page 6, lines 18 onward: these indices should be defined. Alternatively add some references. However, the whole section 2.4 is not very clear. I would suggest another check.
*Down – other references have been added*
Page 7, lines 2-20: I think this part could be shortened again.
*Down – some elements have been deleted*
Page 8, line 5: I would say "346 landslides" instead of "different landslides"
*Down*
Page 9, Table 2: again, landslide type is not a predisposing factor so I suggest removing this row from the table.
*Down*
Page 11, line 6: not clear the meaning of "narrative"
*Down*
Page 13, Figure 3: the sum of the values related to Scenario 2 and 2100 is 100.1%. Please correct.
*Down*
Page 15, lines 12-14: not very clear.
*Down*
Page 27, line 26: "occurrence frequency landslide events" is not clear. Please check.
*Down*
Page 29, line 13: please correct "succion"
*Down*
Page 30, line 14: please correct "It imply"
*Down*
Page 30, line 21: I would say "the evolution of FoS variations range from…"
*Down*
Figures: In some figures, comma is used instead of point as decimal separator (e.g. in the scalebar, 2,5 km). I would suggest correcting it.
*Down in all figures*